# Fractal analysis of geomagnetic data to decipher pre-earthquake process in Andaman-Nicobar region, India

Rahul Prajapati[1*] and Kusumita Arora [1]

[1] Geomagnetism Group, CSIR-National Geophysical Research Institute, Hyderabad-500007, India; rahulphy007@gmail.com

*Correspondence: rahulphy007@gmail.com

**Abstract:** Seismo-electromagnetic (SEM) signatures recorded in geomagnetic data, prior to earthquake, has the potential to reveal pre-earthquake processes in focal zones. The present study analyses the vertical component of geomagnetic field data from Mar 2019 to Apr 2020 using fractal and multifractal approach to identify the EM signatures in Campbell Bay (CBY), a seismically active region of Andaman and Nicobar. The significant enhancements in monofractal dimension and spectrum width components of multifractal analysis arise due to superposition high and low frequency SEM emitted from the pre-earthquake processes. It is observed that the higher frequency components, associated with microfracturing dominate signatures of earthquakes occurring around the West Andaman Fault (WAF) and Andaman Trench (AT), while the lower frequencies, which results from slower electrokinetic mechanisms have some correlation with the earthquakes around the Seulimeum Strand (SS) fault. Thus, the mono fractal, spectrum width, and holder exponent parameter reveals different nature of pre-earthquake processes which can be identified on an average of 10, 12, and 20 days prior to the moderate earthquakes within a radius of 60 km, which holds promise of short -term earthquake prediction.

**Keywords:** Geomagnetic; earthquake precursor; Fractal; Andaman-Nicobar

## 1. Introduction

The existence of precursory signatures prior to an earthquake is a hotly debated topic among researchers across the globe. Several convincing evidences of gas exhalations, variations in groundwater level, temperature variations, fluctuations in the electric and magnetic fields, etc., (Scholz et al., 1973; Rikitake, 1975; Crampin et al., 1980; Bella et al., 1995; Virk et al., 2001; Chadha et al., 2008; Koizumi et al., 2004; Liu et al., 2006; Ouzounov et al., 2007; Panda et al., 1996, 2007; Sethumadhav et al., 2010; Hayakawa and Molchanov, 2004), tilts the scale in favor of detectable signatures of pre-earthquake phenomena. Heterogeneous lithospheric material under strain undergoes micro-fracturing, which causes the polarization of charges, which in turn leads to generation of electromagnetic emission and acousto-gravity waves (Molchanov and Hayakawa, 1995). It has been postulated that most crustal rocks contain dormant electronic charge carriers in the form of peroxy defects, which are released under critical stress levels and flow out of the stressed sub volume as an electric current, which generates magnetic field variations and low frequency EM emissions (Freund and Sornette, 2007). When they reach the Earth's surface, they lead to ionization of air at the ground–air interface (Hayakawa et al., 1996), leading to small disturbances in the local geomagnetic field. Observations of electromagnetic emissions prior to earthquake in frequency ranges from DC, ultra-low frequency, very low frequency, electromagnetic pulses, and very high frequency (Bulusu et al., 2023; Conti et al., 2021; Han et al., 2016; Hattori et al., 2013a; Hayakawa et al., 1999, 1996; Johnston et al., 1984) have been reported by many researchers. Presence of precursory signatures in the ULF range have been extensively studied for earthquakes of M>=7, such as Biak, Spitak, Loma Prieta, Guam, Chi-Chi, Chiapas etc., (Fraser‑Smith et al., 1990; Hattori et al., 2004b; Hayakawa et al., 2000, 1999; Ida et al., 2008; Kopytenko et al., 1993; Molchanov et al., 1992; Smirnova et al., 2013; Stanica and Stănică, 2019; Yen et al., 2004); the ULF range has received more attention as they experience less attenuation and are more likely to reach the Earth's

surface and geomagnetic recording station. Hayakawa et al. (2005) have examined the 3-component data from the same station to identify the anomalous signatures in the polarization ratio of the ULF geomagnetic signal and the diurnal ratio of the Z component for these moderate earthquakes and found a correlatable pattern of these signatures with earthquake occurrence in 75% of the events. This encouraged a deeper investigation into the possible causes of these patterns.

Identification of the geomagnetic anomalies, which are associated with lithospheric processes is a contentious issue. These variations must be uniquely identified, which are distinct from the expressions of magnetospheric-ionospheric processes due to interaction with the solar wind. The most preferred signal processing techniques in previous studies are polarization ratio analysis, diurnal ratio, principal component analysis, singular value decomposition, mono-fractal, and multifractal analysis (Bulusu et al., 2023; Gotoh et al., 2002; Hattori et al., 2004b; Hayakawa et al., 2007, 2005, 1999; Rawat et al., 2016). These signal processing techniques have shown promising results in different cases such as central frequency of 0.01 Hz of non-overlapping window of night time data studied by Han et al. (2015), Hattori et al. (2013b), and Xu et al. (2013), using filtered diurnal signal (using db5 wavelet function) of target station and reference station; Han et al. (2015) have studied diurnal ratio of electric as well as magnetic fields along with polarization ratio of magnetic field of night time data in the ULF range, and Heavlin et al. (2022) studied the signal from a dense network of stations using linear discrimination analysis (LDA) in frequency range 0.001-25 Hz.

The Andaman-Nicobar region lies in the northern part of the Sumatra subduction zone, where the Indian plate is thrusting under the Burma microplate (Gahalaut et al., 2013; Meng et al., 2012; Yang et al., 2017). Persistent tectonic activity is observed here along three major faults, i.e. West Andaman Fault (WAF), Aceh Strands (AS), and Seulimeum Strands (SS). Some of the major earthquakes along these

faults have led to huge losses of life and property and continue to be a worrisome source of mega-scale hazards. During Mar-2019 to Apr-2020, 63 moderate earthquakes of M $\geq$ =4.5 occurred in the vicinity of the geomagnetic station installed by CSIR-NGRI at Campbell Bay (CBY) in Great Nicobar (Figure 1). The property of Self Organized Critically (SOC) of earthquakes provides the motivation to study the fractal characteristics of the geomagnetic time series to decipher the nature of the anomalous signatures in the data (Bak et al., 1988; Hayakawa et al., 1999).

Behavior of natural biological, physical, and geophysical parameters exhibit fractal and multifractal geometries. Mandelbrot (1977, 1982) introduced fractals to characterize the highly complex geometry such as shape of cloud, coastlines, rough surfaces of mountains and landscapes, where traditional Euclidean geometry fails to characterize the nature of such complex geometries, whereas fractals facilitate description of complex geometries (Barnsley et al., 1989). In 1977, after publication of Mandelbrot's book 'Fractals: From, Chance and Dimension', the concept of fractal geometries has been considered as a popular tool among researchers of remote sensing for extraction of land surface features from high resolution remote sense data (Haralick et al. 1973, Weszka et al. 1976, Gong et al. 1992). Several applications of fractals are observed in image processing for decomposition and extraction of image texture (Pentland 1984, Myint 2003). Moreover, the urban system (population size and areas) also shows scaling and SOC nature and the nature of its growth, economics, morphology, genesis and planning well characterize by fractal approach (Keersmaecker et al., 2003; Chen and Zhou, 2008; Chen, 2010). Fractal has diverse application in field of science, such as, medical science (Lopes and Betrouni, 2009), material science (Schafer, 2013), telecommunication (Werner et al., 2002), environmental science (Xu et al., 1993), and computer graphics (Jacquin, 2002). After gaining popularity in space domain, applications of fractal methods on time domain data started in the 1980-s in the field of finance and

economics to characterize rapidly evolving systems. Application of fractals is also observed in geophysical time series data in characterization of natural phenomenon such as solar corona, and space plasmas (Nabulsi and Anukool.,2024; Borovsky, 2021), frequency size distribution of earthquakes or temporal patterns of earthquake parameters such as magnitude, energy, depth, and hypocenter (Hayat et al., 2019; Telesca et al., 2003; Rahimi et al., 2022), and modelling of geological features from geophysical data such as seismology, earthquake dynamics, and well logs etc., (Ahmed et al., 2022; Leary, 1991; Dolan et al., 1988). In recent years, it is noted that, the natural lithospheric processes due tectonic activity such as heat flow on oceanic ridges (Cheng, 2016), mineralization due to hydrothermal (Wang et al., 2017), and earthquakes with different magnitude (Turcotte, 1997) exhibit the fractal nature. From fractal theory, the changes in fractal dimension represent dynamic evolution of the state of the system; the non-linear dynamics of active plate tectonic can be modeled with fractal geometry (Dimri, 2005). The fractal method has become a popular tool in characterization the complexity of dynamic evolution of several type of natural processes including complex behavior of seismicity. The fractal nature of distribution of hypocenter and seismicity pattern was first demonstrated by Kagan and Knopff (1980), and Hirata and Imoto (1991). The spatial distribution of earthquakes shows fractal behavior, wherein the fractal dimension can give an idea of heterogeneities of geological compositions and degree of fracturing of rocks (Pasten and Orrego, 2023). Fractal methods such as Hausdorff dimension, box counting, and correlation dimension are commonly used to study the complex nature of the Earth system and extract deeper insights into seismicity and its relation to tectonic forces (Potirakis et al., 2017; Molchan and Kronrod, 2009; Chen et al., 2006; Mandal et al., 2005). The efficacy of applying the fractal methods to study geomagnetic field patterns prior to earthquake occurrence was a later development (Hattori et al., 2004; Potirakis., 2017; Ida et al., 2012; Hayakawa and Itoh., 2000). For example, in the

case of the Guam earthquake, 1993, a significant change in scaling exponent prior to the event is found (Hayakawa et al., 1999). A similar behavior of scaling exponent was also observed prior to the Biak earthquake in 1996 (Hayakawa et al., 2000).

After the several application of fractals in earthquake research, the researcher found that the earthquake processes and seismicity in time and space are comprises more than one fractal properties i.e. multifractal instead of fractal. Multifractal methods have diverse applications in extracting the dynamic nature of earthquakes in both spatial and time domains. In spatial domain, the multifractal analysis used to characterize the pattern of seismicity, stress distribution, clustering or intermittency of spatial earthquake distribution (Godano et al., 1996; Roy and Mondal, 2012; Casado et al., 2014, Rossi, 1990). Multifractal analysis of the dynamic properties of earthquakes in the time domain reveals the temporal complexity of seismic activity. This insight into earthquake dynamics may aid in forecasting future seismic events. For example, Kiyaschenco et al. (2003) studied the dynamics of seismicity distribution using multifractal parameters (minimum of holder exponent and first order holder exponent) and found a significant decrease prior to major earthquakes. Such characteristics can be used as earthquake precursory signatures. Similarly, Telesca et al. (2004) studied the geomagnetic field from two seismically active regions (Japan and California) and found that temporal variations in multifractal parameters namely entropy and higher-order fractal dimensions, which may indicate processes associated with the preparation of large magnitude earthquakes. Moreover, the generalized multifractal dimension at higher orders (q>1) of ULF geomagnetic field data showed a significant change prior to the 1993 Guam earthquake (Ida et al., 2005). Similarly, multifractal analysis of geomagnetic signals from volcanic eruptions revealed complex dynamics that decreased after eruptions (Currenti et al., 2005). Further, Telesca et al. (2003) analyzed geoelectrical signals recorded in seismically active regions using fractal

and multifractal tools and concluded that the multifractal tools have greater potential for extracting seismo-electrical signatures associated with earthquakes. Smirnova et al. (2013) observed a notable decrease in the higher-order fractal dimension (derived from the generalized fractal dimension) of geomagnetic signals prior to the 1995 Kobe earthquake.

These natural non-linear processes give rise to self-similar pattern and long-range correlations, which are mathematically described by power law relations. Box counting and Hausdorff method are the two fundamentals methods to determine the fractal dimension of geometries in time or space domain. The box counting involves the counting of boxes (of fixed sizes) that contains the at least one values of fractal object (Larry and Toth, 1989). This process is repeated with different box sizes; therefore, the size of boxes and number of boxes with at least one values relate to the fractal dimension of objects. The Hausdorff method is similar to box counting, except that the fractal object is visited by different diameter, and the measured fractal values are called Hausdorff measures. The Hausdorff dimension is related to the Housdorff measures and the variable diameters used for measure the fractal objects. The fractal methods such as Detrended Fluctuation Analysis (DFA), scaling structure function, and Higuchi fractal dimension are common methods for analyzing the geomagnetic signals. Moreover, multifractal geometries do not exhibit self-similar pattern and holding different fractal dimensions. The spectra of fractal dimension values determined from sets of fractals used to delineate the multifractal nature of objects, also known as generalized fractal dimension (Mandelbrot, 1989). In multifractals, the frequency of exponents or fractal dimension indicates the presence of prominent fractal nature of geometries. The strength of fractals or their weight are measured by certain parameter q in the range of $0<q>0$. The multifractal methods, Wavelet Transform Modulus Maxima (WTMM) or wavelet Discrete Wavelet

Transform (DWT), and Multifractal Detrended Fluctuation Analysis (MFDFA) are very common methods for analysis of geomagnetic signals.

For our data, the fractal nature is tested with different approaches (Higuchi, 1988); the Higuchi method provides more consistent and reliable fractal dimension value for the study of fractal behavior of ULF signal (Hattori et al., 2004a; Gotoh et al., 2003; Smirnova et al., 2004). Further, multifractal techniques can better represent the different sources of the signals associated with seismicity (Turcotte, 1989). In this study, we will use nighttime Z-component geomagnetic signal as it is more sensitive to changes in local EM emissions, which are likely to be generated by microfracturing and associated lithospheric deformation. We propose to compute the fractal and multifractal dimensions of the data to extract signatures of more intense perturbations of the signal represented by higher fractal dimension values. The anomalous EM emissions can be correlated with earthquake events in search of pre-earthquake signatures. The earthquake catalog (Table T1) of the study region is adopted from the International Seismological Centre (ISC) with M>= 4.5 and epicenter within 250 km radius of recording station. 63 earthquakes are recorded from 31 March 2019 to 24 April 2020.

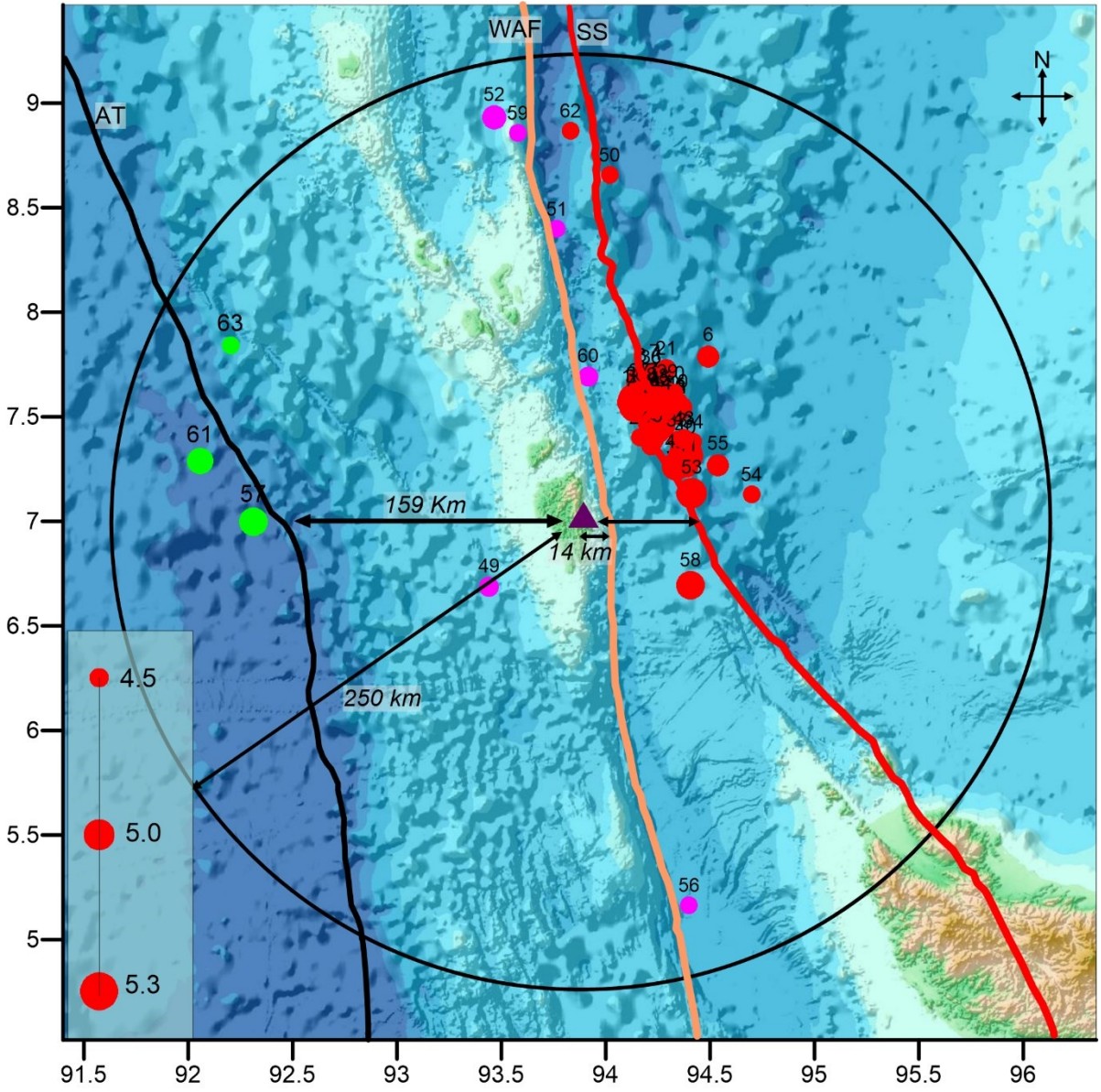

**Figure 1.** Bathymetry map of Andaman-Nicobar subduction zone including Sumatran Fault System; i.e. Seulimeum Strand, West Andaman Fault and Andaman Trench (modified after Cochran 2010; E. Anusha et al., 2020). The circles are representing the earthquake's location and magnitude (size of circle) correspond to each fault system.

## 2. Methodological Approach

It is proposed to apply both fractal and multifractal approaches to the Z component time series, to distinguish between the different source characteristics and examine their relationship to earthquake

parameters. The Z-component of 1 Hz geomagnetic signal analyzed because it is more prone to sense or affected by the local EM field from lithospheric deformation in which vertical components are dominated.

(i) Fractal behavior of Z-component for one-day data using Higuchi is tested and examined. Gotoh et al. (2003) tested different methods for estimation of fractal dimension of geomagnetic signal and suggested that the fractal dimension value using Higuchi method, provided in equation as below, is more reliable and consistent than others. In Higuchi method, a time series $x(n)$ decomposed in to time series of different length $x_k^m$, defined as:

$$x_k^m: x(m), x(m+k), x(m+2k), \ldots\ldots. x\left(m + \left(\frac{N-k}{k}\right).k\right),$$

where, n is 1,2 ,3 …N, $m$ is 1,2,3…$k$, and $k$ is 1,…., $k_{max}$. If the average length of decomposed time series $L_m(k)$ computed at interval of time from $k = 1$ to $k_{max}$ are related to each other as:

$$L(k) \propto k^{-f_D} , \tag{1}$$

then $f_D$ is equal to the slope of fitted line over $\log(L(k))$ versus $\log\left(1/k\right)$ and is considered as fractal dimension of time series data $x(n)$.

The regression line over $\log(L(k))$ versus $\log\left(1/k\right)$ obtained from Higuchi method (indicating power law behaviour) of one-day nighttime (22:00-02:00 LT) Z-component of geomagnetic signal of 3 April 2019, is shown in Figure 2.

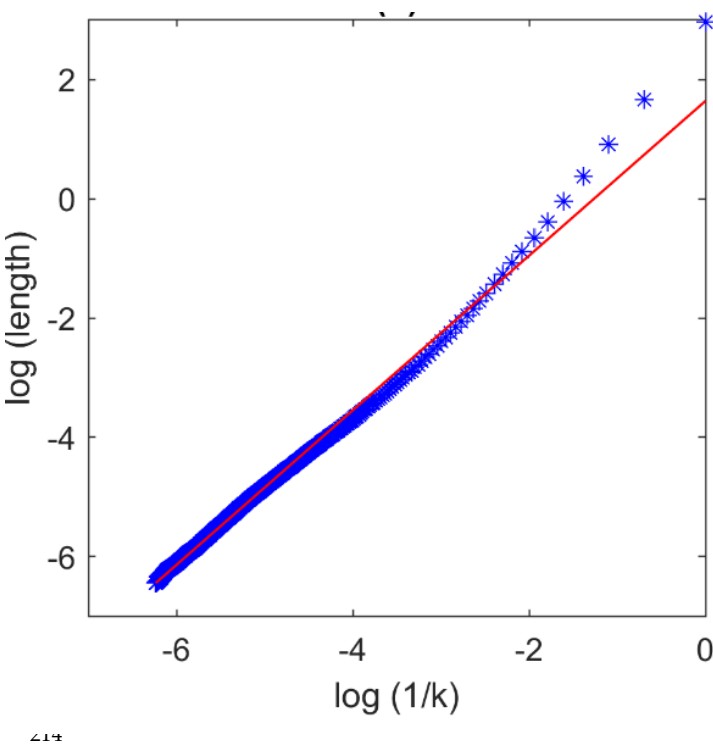

**Figure 2.** The linear fitting over log of average length and log of size of time interval (scale) showing the    215

power law nature of geomagnetic signal.    216

(ii)    For multifractal analyses, the Haar wavelet function is used for discrete wavelet transform because    217

it decomposes the signal into high and low wavelet coefficients. The discrete wavelet transform    218

decomposes the signal up to maximum level defined by $log2(length\ of\ (X(t))/(length\ (\psi_0) + 1)$. The    219

wavelet function $\psi_0$ used to compute the wavelet coefficients of times series $X(t)$ using discrete wavelet    220

transform (DWT) ) with different level of decomposition at dyadic scale $(2^{-j})$ defined as:    221

$$w_x(j,k) = \int X(t)\ 2^{-j}\psi_0(2^{-j}t - k)dt\ ,\qquad\qquad (2)$$    222

where, $w_x(j,k)$ is wavelet coefficients at scale $j$ and time $k$.    223

Further, the wavelet leader values at each level decomposition are defined from $w_x(j,k)$.    224

225

226

227

The wavelet coefficients in dyadic interval $\lambda(j,k)$ at scale $2^j$ is union of two interval at scale $2^{j-1}$, and $3\lambda$ $(j,k)$ is union of three i.e. $\lambda_{j,k-1} \cup \lambda_{j,k} \cup \lambda_{j,k+1}$. Thus, the largest value of coefficients occurred at scale $2^j$ from the union of dyadic scale are referred as wavelet leaders i.e. (Lashermes et al., 2005)

$$L_X(j,k) \equiv L_\lambda = sup_{\lambda' \subset 3\lambda} |w_x(d\lambda')|. \tag{3}$$

Where, $L_X(j,k)$ is wavelet leader at scale $j$ and time $k$.

Since, the time series $X(t)$ hold the condition of regularity, the wavelet leaders follow power law relation and the associated scaling exponent of $X(t)$ at $t0$ is $h(t0)$. The wavelet leaders selected from maximum values of wavelet coefficients at each scale provides the supreme value of scaling exponent i.e. Holder exponent. Thus, the Holder exponent $h$ and wavelet leaders at scale $j$ and time $k$ at limit of fine scales $2^j \rightarrow 0$ are related as (Wendt et al., 2008) i.e.

$$L_X(j,k) \leq C \, 2^{jh}. \tag{4}$$

For the purpose of generalization of Holder exponent values, the structure function of wavelet leader is estimated at each scale $(2^j)$ with moment order $q$. The time averages of (the $q$th powers of) the $L_X(j,k)$ are referred to as the structure functions (with $n_j$) at scale $(2^j)$, which are defined as

$$S^L(q,j) = \frac{1}{n_j} \sum_{k=1}^{n_j} |L_X(j,k)|^q. \tag{5}$$

Where $n_j$ is the number of wavelet leaders at scale j.

Since, the time series function and wavelet leaders hold regularity condition, then the structure functions also follow power law behaviour for $2^j \rightarrow 0$ and can be defined as (Wendt et al., 2007),

$$S^L(q,j) = C_q 2^{j\zeta(q)}. \tag{6}$$

From above relation, the Scaling exponent $\zeta(q)$ are computed from the structure function using regression lines between $log2^j$ versus $S^L(q,j)$, which alternatively can be defined as

$$\zeta_L(q) = \sum_{j=j1}^{2} w_j \, log_2 \, S^L(q,j),$$  (7)

where $w_j$ is weight factor.

Theoretically, the function for multifractal spectrum of Scaling exponent $\zeta_L(q)$ is based on Legendre transforms, defined as

$$f(h) \leq min_{q \neq 0}\big(1 + qh - \zeta L(q)\big),$$  (8)

In the present study, the equations from Wendt et al. (2007) are preferred for the computation of multifractal spectrum from $L_X(j,k)$ i.e.

$$f(q) = \sum_{j=1}^{2} w_j \, U^L(j,q).$$  (9)

$$h(q) = \sum_{j=1}^{2} w_j \, V^L(j,q),$$  (10)

where,

$$U^L(j,q) = \sum_{k=1}^{n_j} R_{X(t)}^q(j,k) \, log_2 R_{X(t)}^q(j,k).$$  (11)

and

$$V^L(j,q) = \sum_{k=1}^{n_j} R_{X(t)}^q(j,k) \, log_2 L_X(j,k),$$  (12)

$$R_{X(t)}^q(j,k) = L_X(j,k)^q \Big/ \sum L_X(j,k)^q.$$  (13)

Larger width of multifractal spectrum indicates larger multifractality or intermittency, and vice-versa. 268

The width of multifractal spectrum $h_w$ (from $-q$ $to$ $+q$) indicates the overall degree of multifractality 269

of signal. The spectrum width $h_{wp}$ ( $q > 0$) and $h_{wn}$ ( $q < 0$) indicates the weaker and stronger 270

singularity of multifractal signal. The $h_{max}$-$h_{min}$ curve defines the average fluctuations embedded in 271

the signal while $h(0)$ represents the zero-order exponent or monofractal dimension (Hayakawa et al., 272

1999). Similarly, $f_{max}$ define the exponent which occurred maximum number of times. Application 273

of multifractal using Haar wavelet on 30 min nighttime (22:00-02:00 LT) data of Z-component of 274

geomagnetic signal of 3 April 2019, is shown in Figure 3. 275

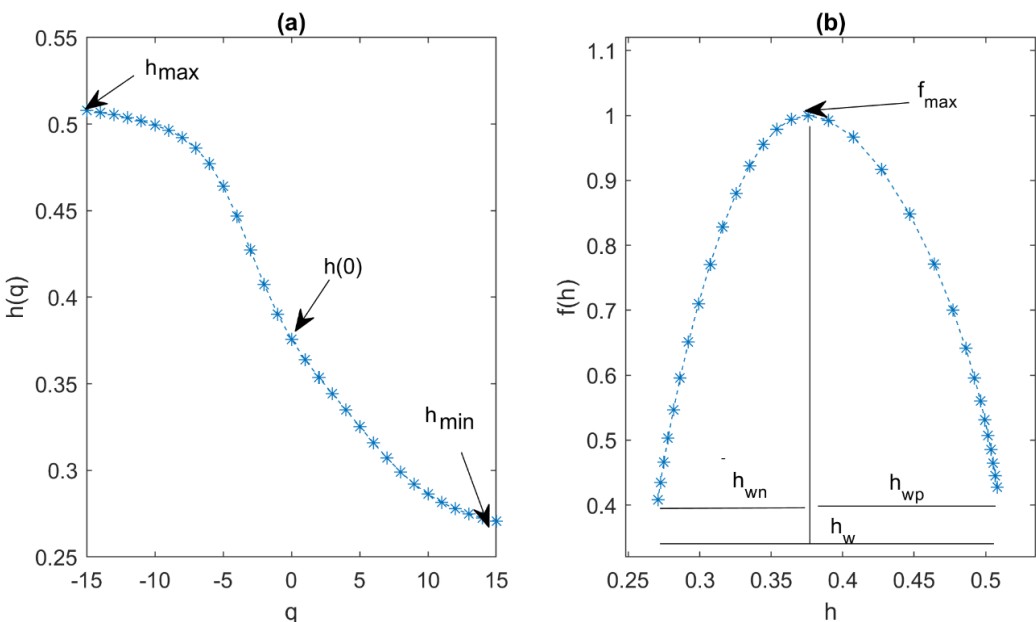

**Figure 3.** The multifractal analysis for 1800 samples of 3$^{rd}$ April 2019, where (a) The variation of holder 285

exponent (h) with moment order q in range of -15 to +15 showing as $h_{min}$, $h_{max}$, and $h(0)$. (b) 286

Multifractal spectrum showing the width of spectrum $h_w$, $h_{wp}$ and $h_{wn}$. 287

288

(i)     The high correlated values measured from fractal, is reason to select the Higuchi method, while for 289

multifractal, wavelet leader is selected due to contact support for wide range of $q$ ($-q$ $to$ $+q$) and 290

stability for scaling function for negative $q$ values compared to other techniques. From fractal, the
power law behaviour, and from multifractal, the finite width of multifractal spectrum and variation
in holder exponent indicates the fractal and multifractal nature of signal, respectively.

(ii)    The fractal dimension $f_D$ of the total duration of Z-component data is calculated for consecutive time
windows of 30 min to trace the variations of the fractal dimension, producing eight values for each
day. The choice of a 30 min time window (consisting of 1800 data points) is based on the balance
between the stability of fluctuations in fractal dimension and minimizing loss of information after
trials with 15 min and 1 hr. time windows.

(iii)   Similarly, the spectrum width parameter ($h_w, h_{wp}, and \ h_{wn}$) and holder exponent parameter $h_{max}$,
$h_{min}$ and, $h(0)$ estimated for the total length of Z component from window of 30 minute to identify
the degree of singularity or complexity (global, weaker, and stronger) as well as degree of
fluctuations with respect to amplitude (from smaller to larger). The shorter fluctuations in fractal
dimensions are smoothed by applying a 15-day moving mean.

(iv)    The increments in fractal dimension and multifractal parameter (spectrum width and holder
exponent) value greater than the threshold value ($\mu + \ \sigma$) are considered as a significant increment as
evidence of existence of EM signatures from lithospheric deformation.

## 3. Results

### 3.1 Monofractal analysis

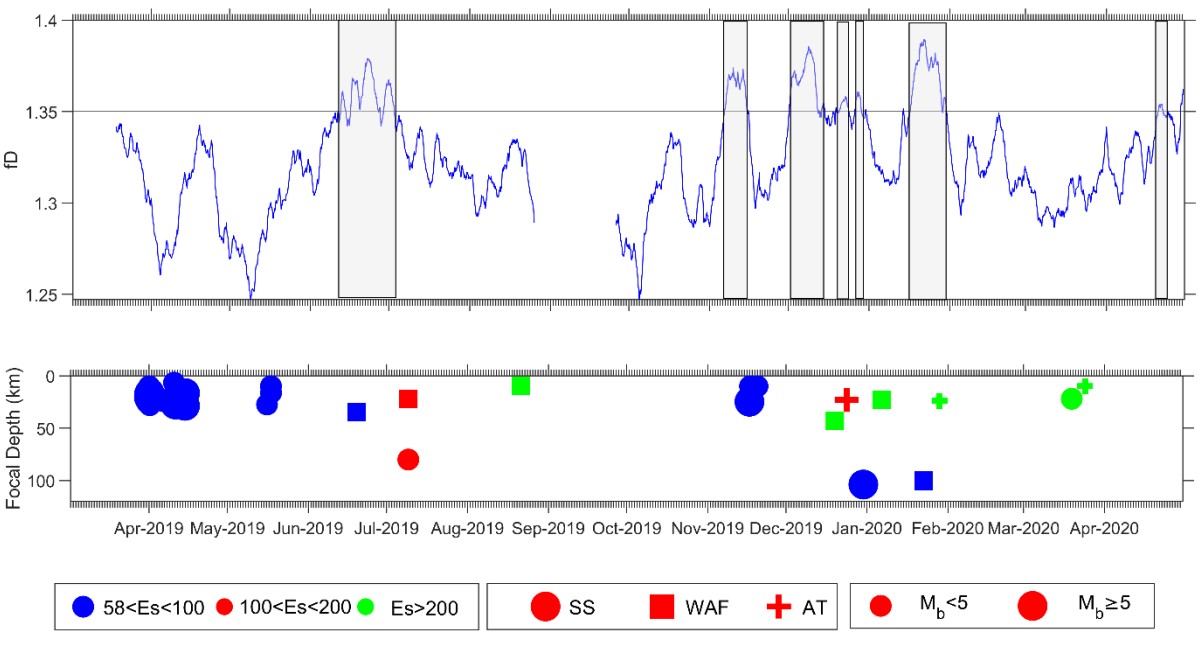

**Figure 4.** (a) Temporal variation of fractal dimension estimated from Higuchi method (15 days moving mean) of Z-component of geomagnetic signal. (b) The time line earthquake occurrences in same duration of geomagnetic signal.

The temporal variations in $f_D$ of vertical component of geomagnetic signal are shown in Figure 4a; $f_D$ greater than the threshold value 1.35 (defined by $\mu + \sigma$ ) are indicated by grey color rectangles. The increasing fractal dimension values are directly proportional to increasing degree of complexity of signal. A synthetic test (supplementary document) of fractal dimension on fraction Brownian motion signals (fBm) with Hurst exponent 02, 0.4, and 0.5 i.e. monofractal signal with increasing degree of complexity (Figure S1) shows higher fractal dimension values (from Higuchi method, Figure S2) for lesser Hurst exponent signal. Moreover, combination of all three signal i.e. multifractal signal shows smaller fractal dimension values indicates that multifractal signal can't be characterized in detail using monofractal dimension. Thus, the observed enhancements in $f_D$ of geomagnetic signal are considered as increasing

complexity from EM signatures caused by impending earthquakes. These enhanced values possibly represent the additional complexity in the signal caused by pre-earthquake microfracturing. The temporal location of enhanced fractal dimension values and their correlations with forthcoming earthquakes are summarized in Table T2. For the earthquake swarm of 1-18 Apr, 2019, and the three earthquakes of 16 & 17th May, 2019, no preceding or coinciding enhancements are recorded. Two phases of enhancements during 12-13 and 16-19 Jun, 2019 occur prior to earthquake of 19th Jun, 2019 (M=4.6 of focal depth of 35 km, along the WAF with epicentral distance of 60 km). The enhancements during 20-26 Jun, and 29 Jun-2 Jul 2019 occur before the dual earthquakes of 9-Jul, 2019 (M=4.5-fd 80 km-epicenter distance 185 km along SS fault; M=4.5-fd 22 km epicenter distance 156 km along WAF). No enhancements beyond threshold value are recorded prior to the very shallow 10 km depth earthquake of 21 Aug (M=4.8) with epicenter 219 km away along the WAF. During Sept and Oct, 2019 neither earthquakes nor enhanced fractal dimensions are observed. Three earthquakes occurred in November, two on 17th and one on the 20th, all on the SS fault. They were of M 5.1, 4.5, 4.7 respectively at shallow focal depths and corresponding epicenters at 60, 91, 78 km from recording site. These events are preceded by a long duration enhancement in fractal dimension from 6-15 Nov. In December, three earthquakes occurred on 19th, 24th and 30th of magnitudes 4.5, 5, 5 respectively on the WAF, AT and SS faults respectively. The earthquakes of 19th Dec of focal depth 43 km and despite large epicentral distance of 212 km from recording site, was preceded by a large amplitude and long duration enhancement of fractal dimension 1-14 Dec; for the next two earthquakes of focal depths 23 and 104 km and corresponding epicentral distances of 173 and 67 km minor enhancements were observed during 18-23 Dec and 26-28 Dec. For the three earthquakes of Jan 2020, the M 4.5 shallow earthquake of 6th Jan with epicentral distance >200 km, no enhancements are observed. The earthquakes of 22nd and 28th Jan occurred. No earthquakes were

recorded in Feb 2020 and no anomalous enhancements are observed. During March 19[th] and 24[th] there were two shallow M=4.5 earthquakes with epicentral distances more than 200 km along the SS and AT respectively. During 20-22 Apr, a small enhancement is observed, the succeeding earthquake in not included in present catalogue.

## 3.2 Multifractal analysis

The holder exponent curve and multifractal spectrum width are calculated for the same data of 3[rd] April, 2019 for the 30 min interval 22:00 – 22:30 LT, with 1800 data points. The large variation in Hurst exponent against moment order $q$ (Figure 4a) and wide width of multifractal spectrum of geomagnetic time series (Figure 4b) indicate the multifractal nature of geomagnetic signal. The multifractal behavior of a signal is generally characterized by the width of multifractal spectrum ($h_w$) as well as spectrum width $h_{wn}$ correspond to $-q$ to 0 and $h_{wp}$ correspond to +q to 0 also assist in characterizing the specific nature of the geomagnetic signal (Figure 4). Apart from spectrum width parameter, holder exponent parameters, such as $h_{min}$, $h_{max}$, $h(0)$, and $f_{max}$ are also useful to characterize the nature of pre-earthquake geomagnetic signal (Figure 4).

## 3.2.1 Multifractal spectrum width

The width of multifractal spectrum deciphers the nature of complexity of analyzed signal; higher spectrum width indicates larger degree of heterogeneity. A synthetic test of multifractal spectrum on fraction Brownian motion signals (fBm) with Hurst exponents 02, 0.4, and 0.5 show increasing width of multifractal spectrum respectively (Figure S3). Moreover, the multifractal spectrum width of combined signal show highest values, indicating increasing nature of complexity, which was not accurately determined by the monofractal dimension. The width of multifractal spectrum ($h_w$, $h_{wp}$ and $h_{wn}$) of a sliding window of 1800 data points (half an hour) without overlapping is computed for whole time series

of vertical component of Z-component (Figure 5). The 15-day moving mean of variation in spectrum width of multifractal spectrum shows significant variations in the range of 0.09 to 0.26. Enhancements greater than threshold value ($\mu + \sigma$) are considered as an anomaly in fractal dimension; . Enhancement in at least one of the components $h_w$, $h_{wp}$ and $h_{wn}$ is considered as significant perturbation of the geomagnetic signal (Figure 5). The enhancements in $h_w$, $h_{wp}$ and $h_{wn}$ components with corresponding earthquakes is summarized in Table T3. For the earthquake swarm of 31 Mar-18 Apr, 2019 (moderate magnitude 4.5-5.3, shallow focal depth 15-30km, and epicentral distance 50-100 km), a preceding enhancement (in $h_w$, $h_{wp}$, and $h_{wn}$) component occurred during 17-22 Mar, 2019. The significant enhancement during 14 May (in $h_w$ component), 14-15 and 17-20 May, 2019 (in $h_{wp}$ component) and 29Apr-5 May, 2019 (in $h_{wn}$ component) are partly common to each other and occurred prior, co and post of earthquake 16$^{th}$ and 17$^{th}$ May, 2019 (moderate magnitude (4.5-4.8), focal depth (10-27.4), and epicentral distance (58-71)). The two sets of enhancement during 22-25 May, 2019 and 4-22 Jun, 2019 (in $h_w$ and $h_{wp}$) and one persistence enhancement during 8-22 Jun, 2019 occurred prior to earthquake 19 Jun, 2019 (M 4.6, focal depth 60 km, and epicentral distance 60 km). the enhancement in common duration 30-9$^{th}$ Jul, 2019 (different duration of persistence) and no enhancement in $h_{wn}$ component occurred prior to two earthquakes 9$^{th}$ Jul, 2019 at two different locations with moderate magnitude (4.5), moderate and shallow focal depth (80 and 22 km) and large epicentral distance (185 and 156 km). The common enhancement during 17-19$^{th}$ Jul, 2019 in $h_w$ and $h_{wn}$ component (not same duration of persistence) occurred prior to earthquake on 21$^{st}$ Aug, 2019 (M 4.8, focal depth 10 km, and large epicentral distance 219 km). the common enhancements during 9-15 Oct, 2019, 7-10$^{th}$ Nov, 2019, in $h_w$ and $h_{wp}$ component, 11-12$^{th}$ Nov in $h_w$, and 2-3, 12-14$^{th}$ Nov, 2019 in $h_{wp}$ component occurred prior to earthquake 17$^{th}$ and 20$^{th}$ Nov, 2019 with moderate magnitude (4.7-5.1), focal depth (10-25 km), and

epicentral distance (60-91 km). Further, the four-earthquake occurred during December, 2019 and 1st<sup></sup> week of Jan, 2020 is not (moderate magnitude, moderate focal depth, and moderate to large epicentral distances) preceded by any significant enhancement in components of multifractal width parameter. The common enhancements during 16-20 Jan, 2020 in $h_w$ and $h_{wp}$ component occurred prior to earthquake 22nd (M 4.6, focal depth 100km, and epicentral distance 77) and 28th Jan, 2020 (M 4.9, focal depth 24km, and epicentral distance 204 km). Further, the two-earthquake event of May-2020 (moderate magnitude, shallow focal depth, and large epicentral distance) is not preceded by any enhancement in components of multifractal width parameter.

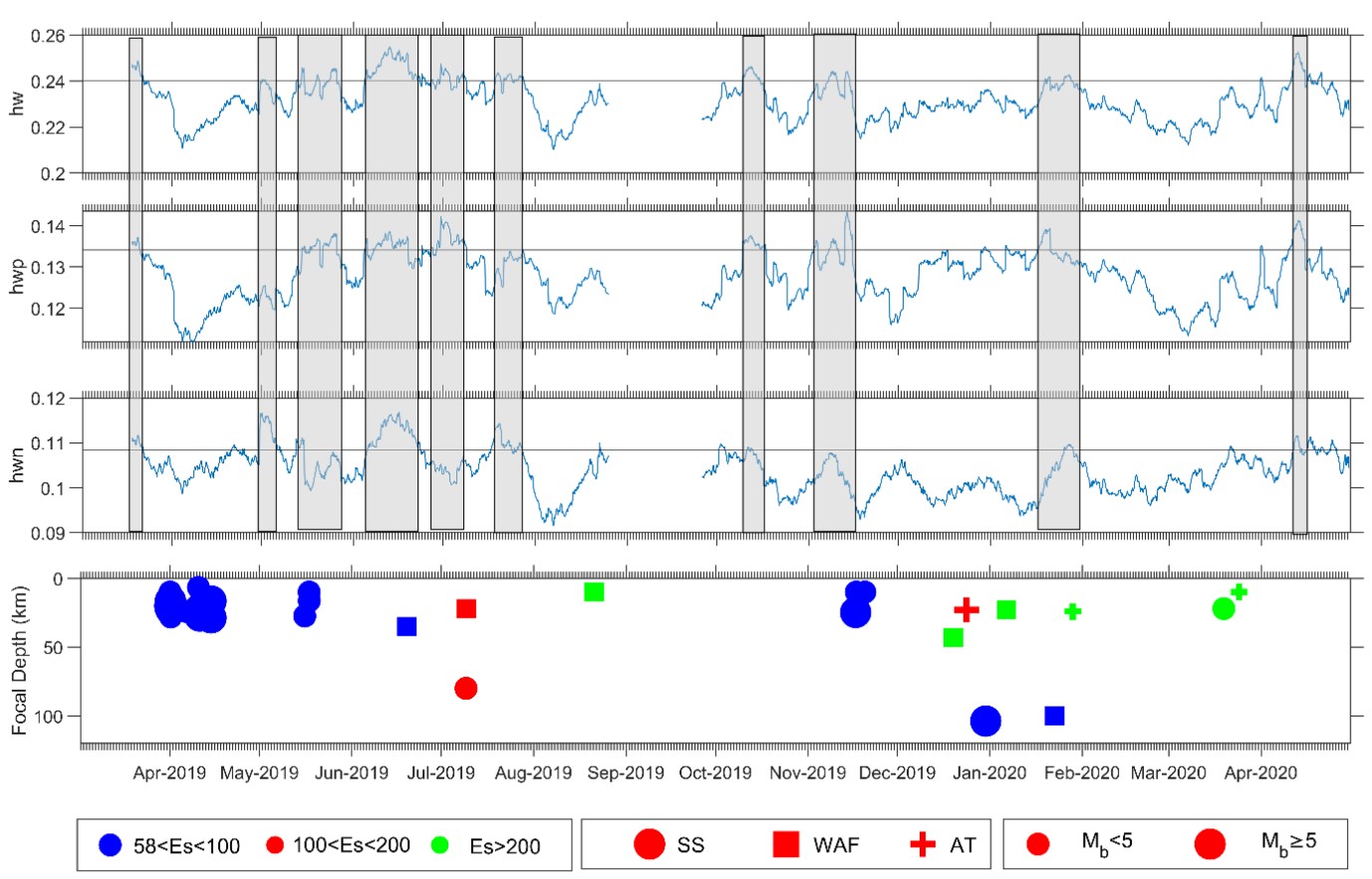

**Figure 5.** Temporal variation in spectrum width $h_w$, $h_{wp}$ and $h_{wn}$ from top panel and anomalous behavior are highlighted by grey color. The bottom panel showing the occurrences of earthquake with magnitude (size of circle) and corresponding faults (different color). Top four panel showing the detail view of Jun 2019 month.

### 3.2.2 Holder Exponent

The holder exponent parameters ($h_{max}$, $h_{min}$, $h(0)$, and $f_{max}$), used for defining the multifractal spectrum curve also show significant variations in the amplitude; again enhancements greater than threshold value (1.0082, 0.4626, 0.5873, 0.3612) are treated as significant (Figure 6). The enhancements in $h_{max}$, $h_{min}$, $h(0)$, and $f_{max}$ components with corresponding earthquakes are summarized in Table

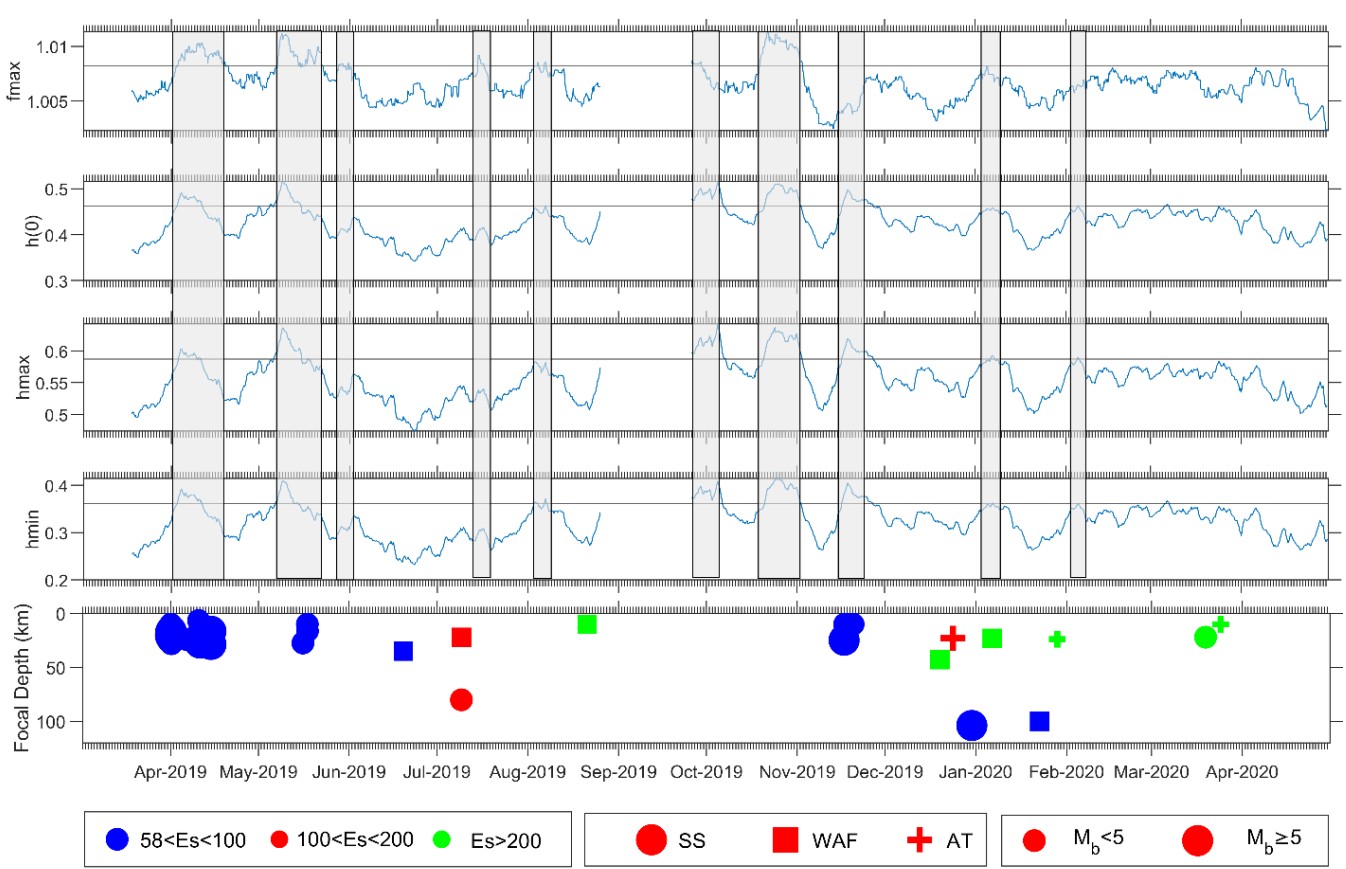

T4.

**Figure 6.** Temporal variation in holder exponent parameters i.e. $f_{max}$ $h_{fmax}$, $h_{max}$ and $h_{min}$ from top panel and anomalous behaviour are highlighted by grey colour. The bottom panel showing the occurrences of earthquake with magnitude (size of circle) and corresponding faults with different color.

The common enhancements during 2-18 April, 2019 in all components of holder exponent coincide with the swarm of earthquake $31^{st}$ $18^{th}$ April, 2019 with moderate magnitude, moderate focal depth, and moderate to large epicentral distance. The next common enhancements are noted during 6-14 May, 2019 in all components of holder exponent prior to the three earthquakes (moderate magnitude, focal depth and epicentral distance), one $16^{th}$ May, 2019, and two $17^{th}$ May, 2019. For the same earthquakes two small co and post seismic enhancements are noted in $f_{max}$ component during 17-19 May, 2019. The small enhancement in only $f_{max}$ during 20-21 May, 2019 is preceded by the earthquake $19^{th}$ Jun, 2019 with moderate magnitude, focal depth, and epicentral distances. Further, the two-earthquake event of $9^{th}$ July with moderate magnitude, epicentral distance, large epicentral distance and different location is not preceded by enhancements in any component of holder exponent. Two small enhancements during 15-16 Jul, and 6 Aug, 2019 in $f_{max}$ component and two small enhancements in $h_{min}$ during 6 Aug, 2019 occurred prior to the earthquake 21 Aug, 2019. The two enhancements common in all components but different durations, one small during 26 Sep-5Oct, 2019 and persistence during 16 Oct-24 Nov, 2019 occurred prior as well as coincident and post three earthquakes. Two of them were at similar location $17^{th}$ Nov, 2019 and one at a different location $20^{th}$ Nov, 2019 with moderate magnitude, shallow to very shallow earthquake, and moderate epicentral distance. Further, the three-earthquake occurred in December, 2019, the first two with moderate magnitude and focal depth and large epicentral distance and third with moderate magnitude, large focal depth, and moderate epicentral distance are not preceded

by enhancement in any component of holder exponent. The next small enhancement in $h_{max}$ component only during 3-8 Jan, 20020 is coincident with earthquake of 06$^{\text{th}}$ Jan, 2020 (mod. Magnitude, mod. Focal depth, and large epicentral distance) and preceded by two earthquakes on 22 and 28$^{\text{th}}$ Jan, 2020 (with moderate magnitude, moderate and large focal depth; large and moderate epicentral distance).

For the earthquake swarm of 31 March, 2019 and early April, the spectrum width shows a small enhancement during 17-20$^{\text{th}}$ March, that is 12 days prior to the earthquake cluster, which have magnitudes between 4.5 to 5.3 and occur in a small region along the SS fault. There is no enhancement of the Holder exponent. For the intermittent earthquakes in mid-April, there is no signal in the spectrum width but the Holder exponent shows a consistent enhance during 3-10 April, a week before the main cluster. In early May, upto 5$^{\text{th}}$, $h_{wn}$ shows an enhancement; the pattern is mimicked in the Holder exponent without crossing the threshold value. Small anomalous enhancements 12-14$^{\text{th}}$ May on the $h_{wn}$, $h_{wp}$ and $h_w$ of spectrum width, just prior to the moderate earthquakes on 16$^{\text{th}}$ and 17$^{\text{th}}$ May. The holder exponent exhibits a longer, more consistent enhancement during 7-14$^{\text{th}}$ May, $f_{max}$ shows a co-seismic anomaly on 17-19 May, followed by anomalies on 20-21 May. Post seismic perturbations are also noted in the spectrum width. For the M4.6 earthquakes of 19$^{\text{th}}$ June, long duration anomalies are seen in spectrum width but not in Holder exponent. For the dual earthquakes on 9$^{\text{th}}$ July, pre and post seismic anomalies are seen in spectrum width; only one anomaly is seen in Holder exponent during 14-16 June. There is no significant multifractal anomaly for the 21 Aug, very shallow earthquake. In October 2019, significant repeated anomalies are observed in Holder exponent right till Nov, 2019. In the second half of Jan and much of February, there are several individual earthquakes; no significant enhancement is observed for any of them. A short enhancement can be noted in 11-14 April, which would be indicative of a future event.

## 3.3 Combined result of monofractal and multifractal analysis

Figures 4, 5, and 6, show that all the components from monofractal and multifractal, have different

response for each earthquake, indicating different characteristics of signal, which can be used as indicator

of pre-earthquake processes in the focal zone of earthquake. In this regard, we have characterized the

enhancements of components in three types of patterns: (i) present in only monofractal component, (ii)

present in only multifractal components, and (iii) present in monofractal as well as in multifractal

component. The significant enhancement from both parameter (monofractal and multifractal) with

corresponding earthquake from figure 4, 5, and 6 is summarized in Figure 7.

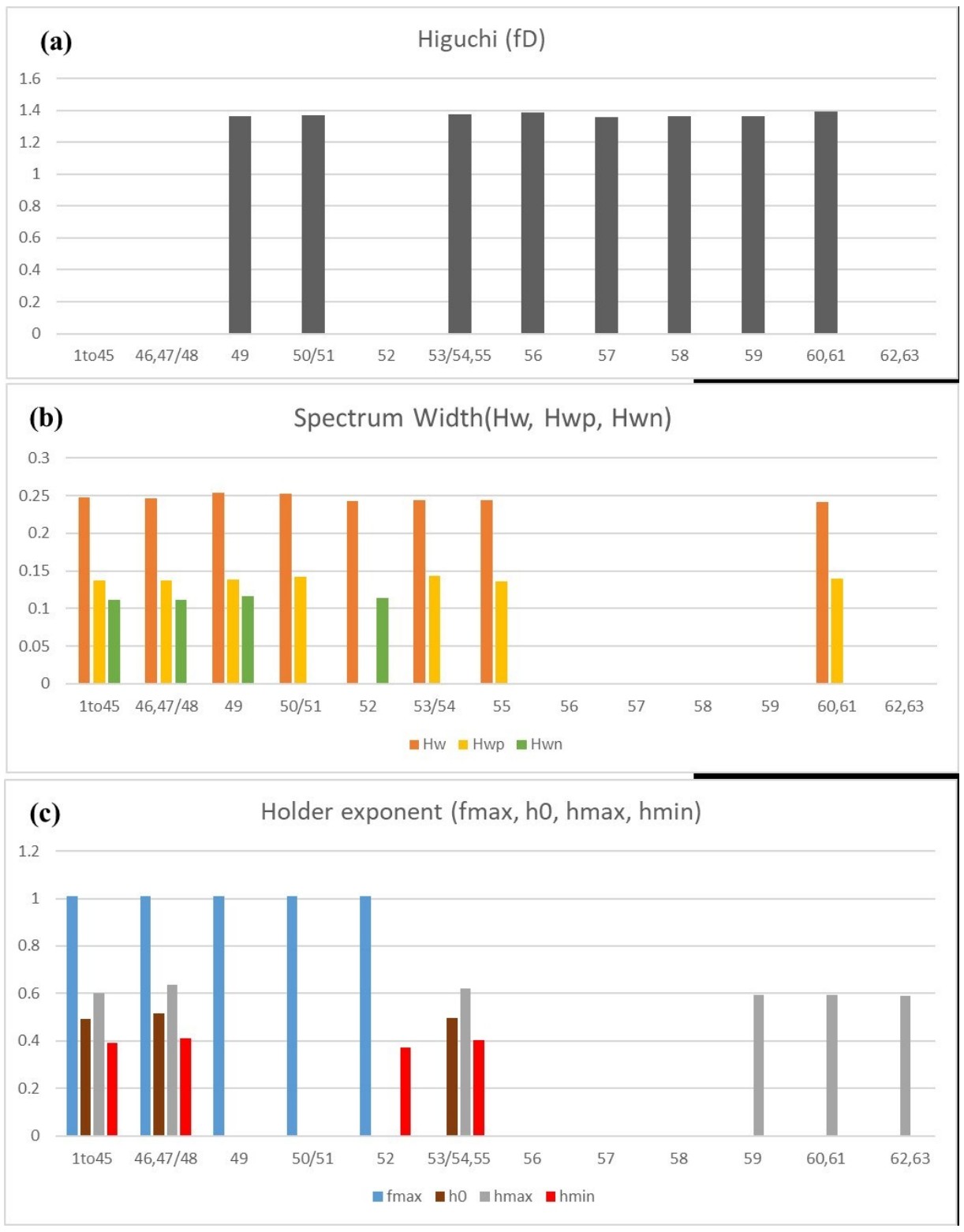

**Figure 7**. The components of significant enhancement with corresponding earthquakes from (a) Higuchi fractal dimension, (b) Spectrum width, and (c) Holder exponent.

From Figure 7 it is evident that the Higuchi fractal dimension from monofractal analysis exhibits

significant enhancements corresponding to earthquake 56, 57, and 58, while there are no enhancements

in multifractal component correspond to same earthquake. Furthermore, there are significant

enhancements in multifractal components correspond to the earthquake 1-45 (swarm of earthquake), 46,

47/48, 52, 62, and 63, while there are no enhancements in monofractal component (or Higuchi fractal

dimension). It is also noted that the earthquake 1-45, 46, 47/48 exhibit to all component of spectrum

width ($h_{wn}$, $h_{wp}$ and $h_w$) and holder exponent $f_{max}, h_{max}, h_{min}, and\ h(0)$, while for earthquake 52

($h_w, h_{wn}, h_{min}, and\ f_{max}$), 62 ($h_{max}$), and 63 ($h_{max}$) all components of multifractal parameters are not

present. Similarly, the significant enhancements correspond to earthquakes 49, 50/51, 53/54, 55, 59, 60,

and 61 observed in monofractal as well as multifractal components, but not in all components of

multifractal.  From multifractal parameters it is also noted that, $h_w$ component of spectrum width is

present in each enhancement, $h_{max}$ component is present with each except for the 49, 50/51, and 52

earthquakes. Similarly, enhancements in $f_{max}$ along with spectrum width $h_w$ is present for all the

earthquakes except 53/54, 55, 60, 61. Significant enhancements for days where the Kp index is greater

than 3 and Dst index smaller than -50 have been identified and removed from the study, although such

short duration effects are diminished considerably after averaging of each component with 15 day

moving mean (Figure 8). An additional component of diurnal ratio is also appended for correlation with

monofractal and multifractal components, which is also treated with criteria of planetary index (figure

8).

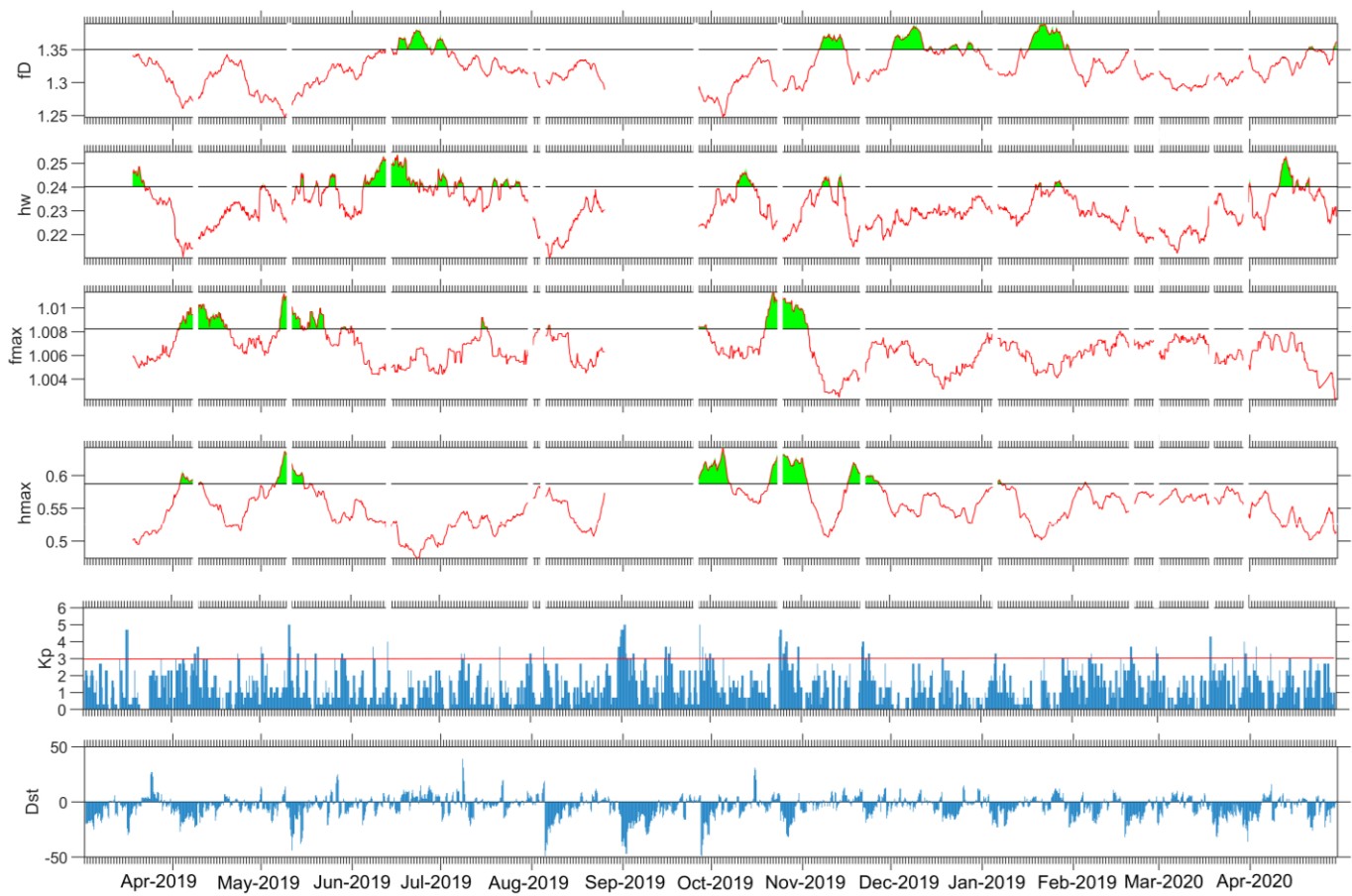

521

522

**Figure 8.** Temporal variation of (a) Higuchi fractal dimension, (b) spectrum width component of multifractal width parameter, (c) fmax component, and (d) hmax component after removing the data correspond to (f) Kp>3 and (g) Dst < -50.

Therefore, from multifractal analysis, $h_w$, $h_{max}$, and $f_{max}$ components, and Higuchi fractal dimension from monofractal parameter has traced all the significant signatures corresponding to the seismogenic activity in the earthquake. The month-wise analysis from Mar-2019 to April -2020 of each component preferred for detail analysis of enhancements shown in Figure S4-S17. From the total duration of analysis, we have selected two quiet days 25th May and 3rd Aug – 2019 and shown the geomagnetic field variation on corresponding date (figure S18), in which first is showing quite disturbed signatures (also showing high multifractal values) compare to second (showing smaller multifractal values). This

suggests that the disturbance in geomagnetic field on the quiet day 25[th] May, 2019 is highly possible due to interference of EM fields.

**Discussion:**

We examine the combined observations of signatures from monofractal or Higuchi fractal dimension ($f_D$) and multifractal components ($h_w$, $h_{max}$ and $f_{max}$) along with diurnal ratio to unravel a linked pattern, which can be interpreted as related to earthquake processes (Figure 9). A swarm of earthquakes (1-45 as per our catalogue) along the SS fault occurred around the first week of April 2019. The data is available from 15[th] March and no anomalies were identified in the Diurnal ratio; hence it was concluded that data length was insufficient (Prajapati and Arora, 2024). While no anomalies were detected in the $f_D$, distinct enhancements are noted in the Spectrum width 14 days prior to the beginning of the swarm. Co-seismic fmax over the entire duration and muted $h_{max}$ enhancements are noted during 2-18 April and 2-10 April respectively.

For the moderate magnitude, shallow focus earthquakes 46, 47, 48, clustered close together during mid-June 2019, Diurnal ratio shows a significant enhancement 50 days before the events, whereas no anomalies are recorded in $f_D$. Enhancements in both hmax and fmax start 11 and 9 days before the events and continue co-seismically.

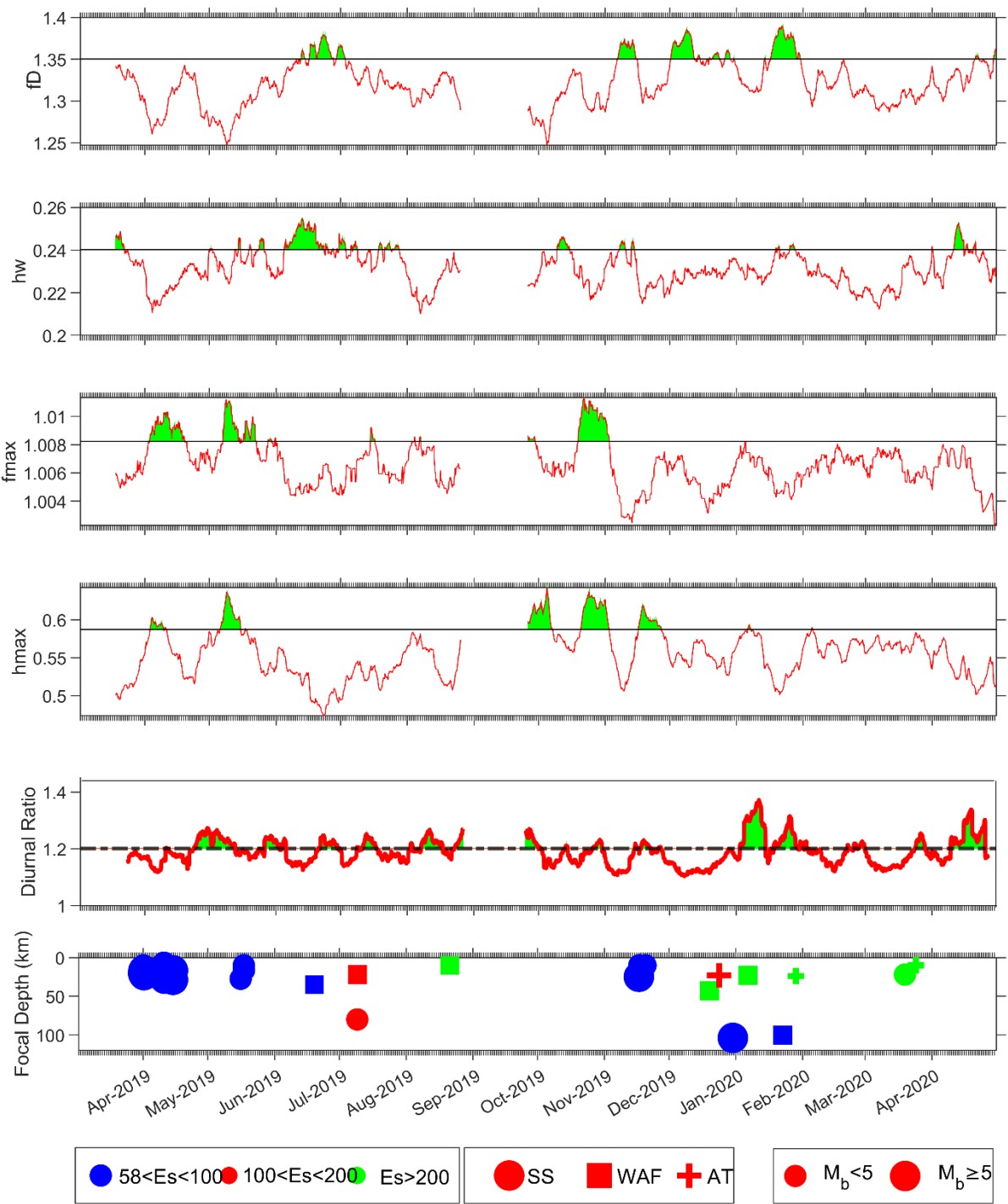

**Figure 9**. The significant enhancement in temporal variation of (a) Higuchi fractal dimension, (b) spectrum width component of multifractal width parameter, (c) fmax component showing the holder exponent presence highest number of time (d) hmax component showing the largest value of holder exponent, and (e) diurnal ratio, indicated by shaded green color, (f) the occurrences of earthquakes in same time duration with magnitude and focal depth.

Earthquake 49 on 19$^{th}$ June 2019 was of moderate magnitude, moderate focal depth and moderate epicentral distance on the WAF. It is preceded by small enhancement in Diurnal ratio 22 days before, $f_D$ 7 days prior and continues co-seismically. Spectrum width enhancement starts 15 days prior to event, which continues co-seismically, there are no signatures in $h_{max}$ or $f_{max}$.

The dual earthquakes 50 and 51, occurred soon after 49, at large epicentral distances on the WAF (shallow focal depth) and on the SS (deep focal depth) in opposite directions to the recording station. Diurnal ratio shows a significant anomaly 16 days prior to the event, accompanied by slight increase in $f_D$ 19 days before. Mild perturbations are also observed in Spectrum width 9-4 days before the events.

The earthquake 52 is similar to 49, with shallower focal depth and very large epicentral distance of 219 km on the WAF. It is preceded by enhancement in Diurnal ratio is seen 14 days before, no signatures are seen in any other parameter.

The earthquakes 53, 54, 55 on 17 and 20 Nov 2019, occur along the SS fault with moderate epicentral distances and shallow focal depth; 53 has magnitude of 5. They are preceded by two phases of small enhancements in Diurnal ratio 21 and 3 days before the earthquakes, continuing to co-seismic signatures. Enhancements in $h_{max}$ continue to co-seismic signatures. Signatures in $h_w$ are very muted, $f_D$ shows significant enhancement 2 days prior to the earthquakes.

Earthquakes 56-63 are individual events, from end of 2019 to first quarter of 2020, separated by several days to weeks intervals in between. Earthquake 56 has very large epicentral distance, also occurring on the WAF like earthquake 52, but with a focal depth of 43 km. This is followed by 57, which is a M=5 earthquake at very shallow focal depth, at large epicentral distance on the AT. Earthquake 58 occurred on Dec 30, 2019, an M=5 event on the SS fault with large focal depth and moderate epicentral distance. The events are preceded by a significant enhancement in $f_D$, but no other signatures. With only one station, it

is not possible to construct an earthquake-anomaly link for this scenario. The cluster of 53-54-55, for which signatures are noted in Diurnal ratio, $f_D$, and $h_{max}$, occurred in a closer duration period, on the same SS fault at moderate epicentral distances and are also at shallow focal depth. The earthquake 59 is of moderate magnitude, shallow focal depth but large epicentral distance on the WAF. Curiously, a co- and post seismic enhancement in diurnal ratio is the sole signature for this event. For the earthquakes 60 (large focal depth and moderate epicentral distance on the WAF) and 61 (shallow focal depth and large epicentral distance on the AT), co-seismic enhancement in diurnal ratio is accompanied by similar enhancement in $f_D$. Earthquakes 62 (moderate magnitude, shallow focal depth and large epicentral distance on the AT) and 63 (moderate magnitude, shallow focal depth and large epicentral distance also on the AT), no preceding signatures are observed on any of the parameters. However, a distinct post seismic increase in diurnal ratio is noted.

In April 2020, enhancements in $h_w$ during 10-14 April and Diurnal ratio during 10-24 April are observed. Several research articles are available (Hayakawa et al., 1999; Gotoh et al., 2003; Ida et al., 2012) to study the behavior of geomagnetic signal using non-linear signal processing techniques such as monofractal and multifractal in context of EM field generated from local sources due to seismogenic activity. Hayakawa et al. (1999) have analysis on H, D, and Z component of ULF geomagnetic signal recorded at 65 km from the epicenter of Guam earthquake (M=8) occurred on 8[th] Oct, 1993 at focal depth of around 60 km carried using fractal (spectral method) and Hurst exponent analysis (rescaled scaled range R/S method). They inferred that decreasing value of slope ($\beta$) from 2.5 to ~1 before the earthquake, which can be considered as an indicator of SOC, where $\beta$ ~1.1 is critical value prior to the earthquake. However, no significant changes observed in Hurst exponent by R/S analysis. The large-scale variation and decrease in ULF spectrum slope (or increase in fractal dimension) means increase high frequency fluctuations is a proxy

measure of small-scale fractal structure cause by active microfracturing process followed by generation of seismogenic ULF emission. In our study, we have also noticed the increase in fractal dimension atleast 10 days prior to the earthquake (49,50-51,53-55, and 60-61) with moderate magnitude (4.5<M<5.1), shallow and moderate focal depth (35, 51,14, and 62km), as well as small, moderate, and large epicentral distance (60, 170, 76, and 140km). The increasing fractal dimension before the earthquakes are suggests the microfracturing process in Earth's crust to be the cause of generation and emission of EM field in the vicinity of recording station.

Gotoh et al. (2003) have analyzed the ULF geomagnetic data recorded at three stations on Izu peninsula, Japan, where a nearby strong earthquake swarm started from 26, June to August 2000 with magnitude upto 6.5. An eruption of volcanic also started simultaneously in Miyakejima Island. Izu region on Philippine plate is under tensile stress and seismically very active because of subduction of Pacific plate at Nankai and Sagami Troughs (Uyeda et al., 2002). The monofractal dimension of the H component shows an increase a week before the earthquake. In present study, we have analyzed Z-component instead of H-component, because recent studies suggested that Z-component is more sensitive for EM fields generated from local sources. In our study we did not find any significant signature of enhanced fractal dimension of Z component one week prior to a swarm of 45 earthquakes from 31-Mar to 18-April, 2019, however an enhancement in spectrum width parameter ($h_w$), 10 days before the swarm activity started.

Further, Ida et al. (2005) carried out the multifractal analysis on H component of geomagnetic signal recorded at 65 km from the epicenter of Guam earthquake occurred on 8[th] Oct, 1993 at focal depth of around 60 km. A westward movement of the Pacific plate and its subduction under Philippine plate triggered the Guam earthquake (Ms 8.0) at shallow dipping subduction zone with a strike slip fault along the trench (Harris, 1993). Ida et al. (2005) found significant changes in the multifractal parameters of Holder exponent

and spectrum width ($\alpha_{min}$, $\alpha_{max}$, $w$, $\Delta$, $f_{max}$, $\alpha$ ($f_{max}$), and $D_q$, for $q < 0, q > 0,$ $and$ $q = 0$). The

observation of 9 days running mean of spectrum width $w$ and $\alpha_{max}$ shows clear and significant variation

30 days prior to the earthquake. In our analysis of multifractal parameters from moderate subduction zone

earthquakes, with focal depth in range of 10-30 km, the 15-day running mean of Spectrum width and Holder

exponent show significant enhancements 12 and 20 days prior to those earthquakes, which occurred close

in time as a cluster (1-45, 47-48, 50-51, 53-55). This difference in pattern may be due to the large

differences in magnitude of the studied earthquakes.

Ida et al. (2012) analyzed the fractal dimension (estimated by Higuchi method) of ULF data recorded at

Kashi station, China, approximately for four years (Mar, 2003 to Dec, 2006), in which several moderate

earthquakes occurred (greater than 5.0 and close to 6) at epicentral distances of 100 to 125, including one

earthquake at approximately 300 km. The region is seismically very active due to relative movement of

plates along SAF fault (normal fault) is locally dominant in the area (He et al., 2015). Ida et al. (2012)

applied the criterion of $\mu \pm 2\sigma$ to define the significant variations of the fractal dimension and reported

decrease in the Z component for two earthquakes (M 5.7 and M 5.4) while the other earthquakes with

magnitude greater than 5 did not show any signature. The enhancement in $f_D$ is interpreted as indication of

dominance of high frequency component and decrease in $f_D$ as dominance of low frequency component,

which may correlate with the high frequency mechanism like micro-fracturing and slow processes like

electrokinetic effect respectively. Potirakis et al.  (2017) has analyzed geomagnetic data (H, D, and Z) at

station Kakioka (KAK) at epicentral distance of 300 km from Tohoku earthquake (M 9.0) of 11 March,

2011. The earthquake was caused by the rupture of a stretch of the subduction zone associated with

the Japan Trench, which separates the Eurasian Plate from the subducting Pacific Plate. The data analyzed

using DFA and Higuchi method, observed a significant decrease in spectral exponent (using DFA) and

corresponding increase in fractal dimension (using Higuchi method) 5-6 months prior to the large 

magnitude Tohoku earthquake. In our study, we have found significant enhancements with the criterion of 

$\mu + \sigma$, producing pre-seismic increases in $f_D$ for multiple earthquake occurrences (50-51, 53-55) with 

4.6<M=5 and either shallow focal depth or small epicentral distance, 19 and 11 days before the earthquakes. 

The concept of self-similarity in time series data was introduced by Mandelbrot and Van Ness (1968) and 

has been used to investigate patterns of seismicity to improve their predictability, as early as the 1990s, e.g. 

Godano and Caruso (1995), who showed that multifractal characteristics of seismic catalogues are more 

appropriate, indicating varying degrees of clustering of seismic events. Fractal analysis has been used to 

study the fractal characteristics of geomagnetic field data to reveal the complexity and irregularity of the 

geomagnetic field, and how it changes in response to different conditions. For example, analysis of the 

fractal properties of the geomagnetic field during different activity levels, showed that the geomagnetic 

field is more multifractal during quiet periods than during storms, and that the scaling properties of the 

field show long-term persistence (Babu and Unnikrishnan, 2023). Another study used the Higuchi 

method to calculate the fractal dimension of the geomagnetic field at a Russian magnetic station and 

found correlations between the fractal dimension and solar wind characteristics and the Auroral Electrojet 

(AE) index (Gvozdarev and Parovik, 2023) and for studying geomagnetic secular variations (Sridharan 

and Ramasamy, 2006). Over the last 20 years many workers have examined the fractal characteristics of 

continuous geomagnetic field data in an earthquake zone to look for indications of anomalous changes in 

fractal dimensions, which may indicate the effect of occurrence of an earthquake. So far the results have 

shown promise, but not yet yielded definitive correlations, a clear argument that many more and systematic 

studies are required. 

Fractal analysis of geomagnetic signals has revealed varying patterns and amplitudes of fractal dimensions representing seismo-electromagnetic (SEM) signatures. The amplitude of enhanced fractal dimension observed by Hayakawa et al. (1999), for a magnitude 8 earthquake is approximately 10 times higher than the fractal dimension observed in our study (for earthquakes of magnitude 4.5-5.1). While enhancements from both studies are linked to microfracturing processes, the variation in amplitude creates ambiguity in connecting parameters such as physical properties of the medium (conductivity, permeability, elastic modulus, etc.), scale of microfracturing, earthquake characteristics (epicentral distance, magnitude, and focal depth), and the method used for computing fractal dimension. Gotoh et al. (2003) observed high fractal dimension values from the H-component (in the noon sector, i.e., 12:00-13:00 LT) as signatures of an earthquake swarm, whereas in our study we found signatures in multifractal parameters of the Z-component (night sector 22:00-02:00 LT. Thus, the fractal dimension shows different results depending on the data component (H or Z) and time of day (day or night) when characterizing similar earthquake events. Ida et al. (2012) observed a decrease in the fractal dimension of the Z-component as a seismic precursor to major earthquakes. This observation contrasts with findings from the 2003 Guam and 2000 Izu Islands earthquake swarms, as well as our studies, which noted an increase in fractal dimension before earthquakes. Ida et al. (2012) suggested that this discrepancy might stem from different dominant processes: inland pre-earthquake activity could be characterized by low-frequency electrokinetic processes, while oceanic activity might be dominated by high-frequency microfracturing processes. It should also be kept in mind that in the tropical regions, any diurnal variation in the atmospheric electrical potential will be more effective to change the electrical current flowing to the Earth's subsurface compared with higher latitudes. Consequently, tectonic faults here can experience greater electrical currents, as increased porosity and micro-fractures make them good conductors. These effects are likely to have a much stronger effect on the

Z component of the geomagnetic field at lower latitudes. Moreover, earthquake catalogs for moderate-magnitude events may offer less precise parameters, such as magnitude, hypocenter, and focal depth. This imprecision can lead to misinterpretation of fractal dimension results in the context of seismo-electromagnetic (SEM) signatures. Thus, interpretations of fractal variations of geomagnetic field data need to be made in the context of earthquake magnitudes and focal depths, focal mechanisms and triggering phenomena, location of the active faults, the distance of the geomagnetic recording station and length of data available, as well as associated EM signatures like TEC changes and radon emissions in a systematic manner, which demand further in-depth study to resolve the ambiguities.

We have defined four clusters of the earthquakes under study (1-45, 47-48, 50-51, 53-55). There are 10 earthquakes, which occurred as single events. For the single events 52, 56-63 (4.5<M<5.0), which are characterized by either large focal depth (>100 km) or large epicentral distance (~200 km), signatures in multifractal parameters. We infer that the EM fields from such moderate magnitude and large epicentral distance earthquakes are too weak to detect by multifractal and diurnal ratio approach (Prajapati and Arora., under review). For the same single events (with focal depth >100km or epicentral distance ~200 km), we observed that enhancements in $f_D$ corresponding to earthquakes 56,57,58, 60, and 61 while the earthquake 52, 59, 62, 63 are not correspond to any pre-co or post enhancements in $f_D$ parameter. The significant enhancement corresponds to 5 events out of 9, including two co-seismic signature (60 and 61) indicate the greater efficacy of $f_D$ parameter than multifractal parameter for single events with focal depth >100km or epicentral distance ~200 km. The earthquake 52 is associated with an increase in the Diurnal ratio 13 days in advance. The single event 49 is characterized by moderate focal depth and epicentral distance, which is associated with co-seismic enhancements in $f_D$, pre-seismic signatures in $h_w$ (7 days prior) and diurnal ratio (15 days prior).

The clusters, on the other hand, produce prominent signatures in the multifractal parameters. The first 

cluster (1-45) has signature in $h_w$ (14 days prior) and a co-seismic enhancement in fmax. The second cluster 

(47-48) has signatures in $f_{max}$, $h_{max}$ and diurnal ratio, 9, 9, 13 days prior to event respectively. The third 

cluster (50-51) at a larger epicentral distance of 165 km, has signatures in $f_D$, $h_w$ and diurnal ratio 19, 9, 

19 days prior to event respectively. The fourth cluster (53-55) includes earthquakes of M=5.1 and the events 

are at shallow focal depth and small-to-moderate epicentral distances produce signatures in $f_D$ and all the 

multifractal parameters as well as diurnal ratio. 

The combined observation from fractal (mono and multifractal) and diurnal ratio (Table 1) clearly indicates 

that the fractal parameters exhibit significant enhancement associated with 10 earthquakes (including co- 

seismic signatures), while significant enhancements in diurnal ratio are correlated with nine earthquakes 

out of ten (including two post-seismic signatures). 

**Table 1:** The following table summarizes the earthquake and its characteristics presence (Y) or absence (-) 

of potential enhancements in monofractal ($f_D$) and multifractal ($h_w$, $f_{max}$, $h_{max}$) components and diurnal 

ratio. 

| EQ. No. | Magnitude | Focal Depth (Km) | Epicentral Distance (Km) | Single (S) /Cluster (C) | $f_D$ | $h_w$ | $f_{max}$ | $h_{max}$ | Diurnal ratio |
|---|---|---|---|---|---|---|---|---|---|
| 1-45 | - | Moderate | Moderate | C | - | Y | Co- | - | - |
| 46-48 | Moderate | Moderate | Moderate | C | - | - | Y | Y | Y |
| 49 | Moderate | Moderate | Moderate | S | Co- | Y | - | - | Y |
| 50-51 | Moderate | Shallow/ Large | Large | C | Y | Y | - | - | Post- |
| 52 | Moderate | Shallow | Large | S | - | - | - | - | Y |
| 53-54- 55 | Large | Shallow | Small | C | Y | Y | Y | Y | Y |

| 56 | Moderate | Moderate | Large | S | Y | - | - | - | - |
|---|---|---|---|---|---|---|---|---|---|
| 57 | Large | Shallow | Large | S | Y | - | - | - | - |
| 58 | Large | Large | Mod | S | Y | - | - | - | - |
| 59 | Moderate | Shallow | Large | S | - | - | - | - | Y |
| 60 | Moderate | Large | Moderate | S | Co- | - | - | - | Y |
| 61 | Moderate | Shallow | Large | S | Co- | - | - | - | Y |
| 62 | Moderate | Shallow | Large | S | - | - | - | - | - |
| 63 | Moderate | Shallow | Large | S | - | - | - | - | post |

According to Ida et al. (2012), significant enhancements in fractal values of geomagnetic signal recorded in tectonic active areas are representing the dominance of high frequency component associated with EM field from microfracturing processes in lithosphere. Apart from this, the components of holder exponent (part of multifractal analysis) such as $f_{max}$ $h_{max}$, $h_{min}$, and $h(0)$ also analyses the different characteristics of the signal (Krzyszczak et al., 2019) such as enhancement in $h_{max}$ indicates that underlying process of events are more smooth rather than sorter fluctuations while $h_{min}$ is just opposite to $h_{max}$. Similarly, $f_{max}$ is correspond to $h0$ i.e. $h$ which occurred maximum number of times in range $h_{max}$- $h_{min}$. The enhancements in $f_{max}$ value with large $h$ indicate the underlying processes is less correlated and fine structure i.e. signal embedded with anomalies and not completely regular while $f_{max}$ correspond to smaller value of $h$ indicate the highly correlated and most regular signal. Enhancements in $h_{max}$ and $f_{max}$ with $h0$ correspond to large $h$ of a geomagnetic signal recorded in active tectonic area, indicates that the underlying processes is smooth and exhibit anomalies (less correlated and fine structures) of low frequencies. According to Conti et al. (2021) electrokinetic process is responsible for generation of low frequency EM signature from lithospheric deformation of a focal zone.

The enhancements in $h_{max}$ and $f_{max}$, preceding the clusters of shallow earthquakes 1-45, 46-48, 53-55 on the SS fault at moderate epicentral distances are indicative of low frequency perturbations from multiple

sources, which are ascribed to electrokinetic processes (Conti et al., 2021). For the cluster 50-51, the former

occurs on the SS fault and the latter on the WAF leading to interferences of the EM signals, whereby the

$h_{max}$ and $f_{max}$

enhancements are not prominent.

 The earthquakes 49, 51 and 52 on the WAF dominated by strike slip mechanisms are also shallow and are

at moderate epicentral distances but have enhancements in $f_D$ and $h_w$, the latter being more significant.

This is interpreted as high frequency perturbations attributed to microfracturing processes (Ida et al., 2012).

The earthquakes 56, 57, 59, 60, 61, 63 on the WAF and AT faults at large epicentral distances are linked

with enhancements in $f_D$ and $h_w$, the former being more significant. We interpret these high frequency

perturbations to be also generated due to microfracturing processes; the large epicentral distances possibly

leading to attenuation of the highest frequency components leads to more prominent monofractal

signatures. The earthquakes 50, 58 and 62 are either at very large epicentral distances or large focal depths

and fail to produce signatures in any of the fractal components.

Thus, the moderate focal depth and epicenter distance earthquakes on WAF are dominated by $h_w$ while

large focal depth and epicentre distance earthquakes on WAF/AT dominated by $f_D$ possibly indicating that

the EM field from large distance are more homogeneous due to attenuation and dominating its appearance

in $f_D$ component, while EM field from short distance, indicating that EM field are more heterogeneous and

dominating its appearance in $h_w$ component. Which means, $f_D$ component is most sensitive component for

large epicenter and focal depth earthquakes while $h_w$ component is more sensitive for moderate epicentre

distance and focal depth earthquakes.

**5. Conclusions**

The study of fractal natures of the geomagnetic time series (Z component) allows us to conclude:

(i) The earthquake clusters occurred on normal/thrust fault are of moderate magnitude and focal depth are emitting prior EM fields of low frequency effectively generated from electrokinetic processes in focal zone of earthquake.

(ii) The single earthquakes occurred on strike slip WAF fault of moderate magnitude and focal depth are emitting prior EM field of more heterogeneity and frequency while, earthquakes on same fault with large epicentre distance/ focal depth emitting prior EM field of lesser heterogeneity and high frequency effectively generated from microfracturing processes in focal zone of earthquake.

(iii) The monofractal dimension $f_D$ is more effective to trace the EM field from large epicentre distance and focal depth while multifractal spectrum width $h_w$ is more effective to trace the EM field from moderate to small epicentre distance and focal depth for the case of microfracturing processes.

(iv) The fractal analysis has advantage over diurnal ratio is simultaneous observation of high and low frequency EM field from lithospheric deformation of focal zone of earthquake, which are emitted from different pre-earthquake processes.

## Statements and Declarations

### (i) Data Availability

The data that support the findings of this study are available upon reasonable request.

### (ii) Competing Interests

The authors have no relevant financial or non-financial interests to disclose.

### (iii) CRediT authorship contribution statement

All authors contributed to the study conception and design. Methodology and data collection were performed by Kusumita Arora, and Rahul Prajapati. Data curation and its analysis using MATLAB coding was performed by Rahul Prajapati. The first draft of the manuscript was written by Rahul Prajapati. Review and editing of first draft of the manuscript performed by

Kusumita Arora, and the work carried out under supervision and validation of Kusumita Arora. 811
All authors read and approved the final manuscript. 812

813

**Acknowledgments:** The Authors are thankful to the Director CSIR-National Geophysical Research 814
Institute, India for granting permission to access the data for research purpose and to publish the work 815
(Ref. No. NGRI/Lib/2024/Pub-019). The authors acknowledge the available public domain data sets 816
from WDC Kyoto (https://wdc.kugi.kyoto-u.ac.jp/) and earthquake data from ISC catalogue 817
(http://www.isc.ac.uk/iscbulletin/search/catalogue/). Authors are also acknowledging the Dr. N. Phani 818
Chandrasekhar and other observatories staff for maintaining the remote site observatories to acquire the 819
uninterrupted data. 820

821

**References** 822

Babu, S. S. and Unnikrishnan, K.: Analysis of fractal properties of horizontal component of Earth's 823
magnetic field of different geomagnetic conditions using MFDFA, Adv. Sp. Res., 72, 2391–2405, 2023. 824

825

Bak, P., Tang, C., and Wiesenfeld, K.: Self-organized criticality, Phys. Rev. A, 38, 364, 1988. 826

Barnsley, M. F., Elton, J., Hardin, D., and Massopust, P.: Hidden variable fractal interpolation functions, 827
SIAM J. Math. Anal., 20, 1218–1242, 1989. 828

Barabási, A.-L. and Vicsek, T.: Multifractality of self-affine fractals, Phys. Rev. A, 44, 2730, 1991. 829
Borovsky, J. E.: Magnetospheric plasma systems science and solar wind plasma systems science: The 830
plasma-wave interactions of multiple particle populations, Front. Astron. Sp. Sci., 8, 780321, 2021. 831

Bella, J., Brodsky, B., and Berman, H. M.: Hydration structure of a collagen peptide, Structure, 3, 893– 832
906, 1995. 833

Bhattacharya, K. and Manna, S. S.: Self-organized critical models of earthquakes, Phys. A Stat. Mech. its 834
Appl., 384, 15–20, 2007. 835

Bulusu, J., Arora, K., Singh, S., and Edara, A.: Simultaneous electric, magnetic and ULF anomalies 836
associated with moderate earthquakes in Kumaun Himalaya, Nat. Hazards, 1–31, 2023. 837

Borovsky, J. E. and Valdivia, J. A.: The Earth's magnetosphere: a systems science overview and assessment, Surv. Geophys., 39, 817–859, 2018.

Chadha, R. K., Singh, C., and Shekar, M.: Transient changes in well-water level in bore wells in Western India due to the 2004 M W 9.3 Sumatra Earthquake, Bull. Seismol. Soc. Am., 98, 2553–2558, 2008.

Chen, C. C., Wang, W. C., Chang, Y. F., Wu, Y. M., and Lee, Y. H.: A correlation between the b-value and the fractal dimension from the aftershock sequence of the 1999 Chi-Chi, Taiwan, earthquake, Geophys. J. Int., 167, 1215–1219, https://doi.org/10.1111/j.1365-246X.2006.03230.x, 2006.

Chen, Y.: Characterizing growth and form of fractal cities with allometric scaling exponents, Discret. Dyn. Nat. Soc., 2010, https://doi.org/10.1155/2010/194715, 2010.

Chen, Y. and Zhou, Y.: Scaling laws and indications of self-organized criticality in urban systems, Chaos, Solitons and Fractals, 35, 85–98, https://doi.org/10.1016/j.chaos.2006.05.018, 2008.

Conti, L., Picozza, P., and Sotgiu, A.: A critical review of ground based observations of earthquake precursors, Front. Earth Sci., 9, 676766, 2021.

Currenti, G., Del Negro, C., Lapenna, V., and Telesca, L.: Natural Hazards and Earth System Sciences Multifractality in local geomagnetic field at Etna volcano, Sicily (southern Italy), Natural Hazards and Earth System Sciences, 555–559 pp., 2005.

Crampin, S., McGonigle, R., and Bamford, D.: Estimating crack parameters from observations of P-wave velocity anisotropy, Geophysics, 45, 345–360, 1980.

Currenti, G., Del Negro, C., Lapenna, V., and Telesca, L.: Multifractality in local geomagnetic field at Etna volcano, Sicily (southern Italy), Nat. Hazards Earth Syst. Sci., 5, 555–559, 2005.

Dimri, V. P.: Fractal behaviour of the earth system, Springer, 2005.

El-Nabulsi, R. A. and Anukool, W.: Time-dependent heating problem of the solar corona in fractal dimensions: A plausible solution, Adv. Sp. Res., 74, 2510–2529, https://doi.org/10.1016/j.asr.2024.06.015, 2024.

Fraser-Smith, A. C., Bernardi, A., McGill, P. R., Ladd, M., Helliwell, R. A., and Villard Jr, O. G.: Low-frequency magnetic field measurements near the epicenter of the Ms 7.1 Loma Prieta earthquake, Geophys. Res. Lett., 17, 1465–1468, 1990.

Freund, F. and Sornette, D.: Electro-magnetic earthquake bursts and critical rupture of peroxy bond

networks in rocks, Tectonophysics, 431, 33–47, 2007.

Gahalaut, V. K., Kundu, B., Laishram, S. S., Catherine, J., Kumar, A., Singh, M. D., Tiwari, R. P., Chadha, R. K., Samanta, S. K., and Ambikapathy, A.: Aseismic plate boundary in the Indo-Burmese wedge, northwest Sunda Arc, Geology, 41, 235–238, 2013.

Gao, X.-Y., Guo, Y.-J., and Shan, W.-R.: Optical waves/modes in a multicomponent inhomogeneous optical fiber via a three-coupled variable-coefficient nonlinear Schrödinger system, Appl. Math. Lett., 120, 107161, 2021.

Godano, C., Alonzo, M. L., and Bottari, A.: Multifractal analysis of the spatial distribution of earthquakes in southern Italy, Geophys. J. Int., 125, 901–911, https://doi.org/10.1111/j.1365-246X.1996.tb06033.x, 1996.

Gong, P. and Howarth, P. J.: The use of structural information for improving land-cover classification accuracies at the rural-urban fringe, Photogramm. Eng. Remote Sens., 1990.

Gotoh, K., Akinaga, Y., Hayakawa, M., and Hattori, K.: Principal component analysis of ULF geomagnetic data for Izu islands earthquakes in July 2000, J. Atmos. Electr., 22, 1–12, 2002.

Gotoh, K., Hayakawa, M., and Smirnova, N.: Fractal analysis of the ULF geomagnetic data obtained at Izu Peninsula, Japan in relation to the nearby earthquake swarm of, Natural Hazards and Earth System Sciences, 229–236 pp., 2003.

Gotoh, K., Hayakawa, M., Smirnova, N. A., and Hattori, K.: Fractal analysis of seismogenic ULF emissions, Phys. Chem. Earth, Parts A/B/C, 29, 419–424, 2004.

Gvozdarev, A. and Parovik, R.: On the Relationship between the Fractal Dimension of Geomagnetic Variations at Altay and the Space Weather Characteristics, Mathematics, 11, 3449, https://doi.org/10.3390/math11163449, 2023.

Han, P., Hattori, K., Xu, G., Ashida, R., Chen, C.-H., Febriani, F., and Yamaguchi, H.: Further investigations of geomagnetic diurnal variations associated with the 2011 off the Pacific coast of Tohoku earthquake (Mw 9.0), J. Asian Earth Sci., 114, 321–326, 2015.

Han, P., Hattori, K., Huang, Q., Hirooka, S., and Yoshino, C.: Spatiotemporal characteristics of the geomagnetic diurnal variation anomalies prior to the 2011 Tohoku earthquake (Mw 9.0) and the possible coupling of multiple pre-earthquake phenomena, J. Asian Earth Sci., 129, 13–21, 2016.

Haralick, R. M., Shanmugam, K., and Dinstein, I. H.: Textural features for image classification, IEEE Trans. Syst. Man. Cybern., 610–621, 1973.

Harris, S. K.: NATIONAL CENTER FOR EARTHQUAKE The Island of Guam Earthquake of, 1993.

Hattori, K., Serita, A., Gotoh, K., Yoshino, C., Harada, M., Isezaki, N., and Hayakawa, M.: ULF geomagnetic anomaly associated with 2000 Izu islands earthquake swarm, Japan, Phys. Chem. Earth, Parts

A/B/C, 29, 425–435, 2004a.

Hattori, K., Takahashi, I., Yoshino, C., Isezaki, N., Iwasaki, H., Harada, M., Kawabata, K., Kopytenko, E., Kopytenko, Y., Maltsev, P., Korepanov, V., Molchanov, O., Hayakawa, M., Noda, Y., Nagao, T., and Uyeda, S.: ULF geomagnetic field measurements in Japan and some recent results associated with Iwateken Nairiku Hokubu earthquake in 1998, Phys. Chem. Earth, 29, 481–494, https://doi.org/10.1016/j.pce.2003.09.019, 2004b.

Hattori, K.: ULF geomagnetic changes associated with large earthquakes, Terr. Atmos. Ocean. Sci., 15, 329–360, 2004.

Hayakawa, M., Ito, T., and Smirnova, N.: Fractal analysis of ULF geomagnetic data associated with the Guam earthquake on August 8, 1993, Geophys. Res. Lett., 26, 2797–2800, https://doi.org/10.1029/1999GL005367, 1999.

Hayakawa, M., Itoh, T., Hattori, K., and Yumoto, K.: ULF electromagnetic precursors for an earthquake at Biak, Indonesia on February 17, 1996, Geophys. Res. Lett., 27, 1531–1534, https://doi.org/10.1029/1999GL005432, 2000.

Hayat, U., Barkat, A., Ali, A., Rehman, K., Sifat, S., and Iqbal, T.: Fractal analysis of shallow and intermediate-depth seismicity of Hindu Kush, Chaos, Solitons and Fractals, 128, 71–82, https://doi.org/10.1016/j.chaos.2019.07.029, 2019.

Hattori, K., Han, P., Yoshino, C., Febriani, F., Yamaguchi, H., and Chen, C. H.: Investigation of ULF Seismo-Magnetic Phenomena in Kanto, Japan During 2000-2010: Case Studies and Statistical Studies, https://doi.org/10.1007/s10712-012-9215-x, 1 May 2013a.

Hattori, K., Han, P., Yoshino, C., Febriani, F., Yamaguchi, H., and Chen, C.-H.: Investigation of ULF seismo-magnetic phenomena in Kanto, Japan during 2000–2010: case studies and statistical studies, Surv. Geophys., 34, 293–316, 2013b.

Hayakawa, M. and Molchanov, O. A.: Summary report of NASDA's earthquake remote sensing frontier project, Phys. Chem. Earth, Parts A/B/C, 29, 617–625, 2004.

Hayakawa, M., Kawate, R., Molchanov, O. A., and Yumoto, K.: Results of ultra-low-frequency magnetic field measurements during the Guam earthquake of 8 August 1993, Geophys. Res. Lett., 23, 241–244, 1996.

Hayakawa, M., Ito, T., and Smirnova, N.: Fractal analysis of ULF geomagnetic data associated with the Guam earthquake on August 8, 1993, Geophys. Res. Lett., 26, 2797–2800, https://doi.org/10.1029/1999GL005367, 1999.

Hayakawa, M., Itoh, T., Hattori, K., and Yumoto, K.: ULF electromagnetic precursors for an earthquake at Biak, Indonesia on February 17, 1996, Geophys. Res. Lett., 27, 1531–1534, 2000.

Hayakawa, M., Ida, Y. U. I., and Gotoh, K.: Multifractal analysis for the ULF geomagnetic data during the Guam earthquake, in: IEEE 6th International Symposium on Electromagnetic Compatibility and Electromagnetic Ecology, 2005, Proceedings, 239–243, https://doi.org/10.1109/EMCECO.2005.1513113, 2005.

Hayakawa, M., Hattori, K., and Ohta, K.: Monitoring of ULF (ultra-low-frequency) Geomagnetic Variations Associated with Earthquakes, Sensors, 7, 1108–1122, 2007.

He, P., Wen, Y., Xu, C., Liu, Y., and Fok, H. S.: New Evidence for Active Tectonics at the Boundary of the Kashi Depression , China , from Time Series InSAR Observations Tectonophysics New evidence for active tectonics at the boundary of the Kashi Depression , China , from time series InSAR observations, Tectonophysics, 653, 140–148, https://doi.org/10.1016/j.tecto.2015.04.011, 2015.

Heavlin, W. D., Kappler, K., Yang, L., Fan, M., Hickey, J., Lemon, J., MacLean, L., Bleier, T., Riley, P., and Schneider, D.: Case-Control Study on a Decade of Ground-Based Magnetometers in California Reveals Modest Signal 24–72 hr Prior to Earthquakes, J. Geophys. Res. Solid Earth, 127, https://doi.org/10.1029/2022JB024109, 2022.

Higuchi, T.: Approach to an irregular time series on the basis of the fractal theory, Phys. D Nonlinear Phenom., 31, 277–283, 1988.

Hirata, T. and Imoto, M.: Multifractal analysis of spatial distribution of microearthquakes in the Kanto region, Geophys. J. Int., 107, 155–162, 1991.

Ida, Y., Hayakawa, M., Adalev, A., and Gotoh, K.: Multifractal analysis for the ULF geomagnetic data during the 1993 Guam earthquake, Nonlinear Process. Geophys., 12, 157–162, https://doi.org/10.5194/npg-12-157-2005, 2005.

Ida, Y., Yang, D., Li, Q., Sun, H., and Hayakawa, M.: Detection of ULF electromagnetic emissions as a precursor to an earthquake in China with an improved polarization analysis, Hazards Earth Syst. Sci, 775–777 pp., 2008.

Ida, Y., Yang, D., Li, Q., Sun, H., and Hayakawa, M.: Fractal analysis of ULF electromagnetic emissions in possible association with earthquakes in China, Nonlinear Process. Geophys., 19, 577–583, https://doi.org/10.5194/npg-19-577-2012, 2012.

Jacquin, A. E.: Fractal image coding: A review, Proc. IEEE, 81, 1451–1465, 1993.

Jaffard, S., Lashermes, B., and Abry, P.: Wavelet leaders in multifractal analysis, Wavelet Anal. Appl., 1, 219–264, 2006.

Johnston, M. J. S., Mueller, R. J., Ware, R. H., and Davis, P. M.: Precision of geomagnetic field measurements in a tectonically active region, J. Geomagn. Geoelectr., 36, 83–95, 1984.

Kagan, Y. Y. and Knopoff, L.: Spatial distribution of earthquakes: the two-point correlation function,

Geophys. J. Int., 62, 303–320, 1980.

Kantelhardt, J. W., Zschiegner, S. A., Koscielny-Bunde, E., Havlin, S., Bunde, A., and Stanley, H. E.: Multifractal detrended fluctuation analysis of nonstationary time series, Phys. A Stat. Mech. its Appl., 316, 87–114, 2002.

Keersmaecker, De. M. L., Frankhauser, P., and Thomas, I.: Using fractal dimensions for characterizing intra-urban diversity: The example of Brussels, Geogr. Anal., 35, 310–328, https://doi.org/10.1111/j.1538-4632.2003.tb01117.x, 2003.

Kiyashchenko, D., Smirnova, N., Troyan, V., and Vallianatos, F.: Dynamics of multifractal and correlation characteristics of the spatio-temporal distribution of regional seismicity before the strong earthquakes, Natural Hazards and Earth System Sciences, 285–298 pp., 2003.

Koizumi, N., Kitagawa, Y., Matsumoto, N., Takahashi, M., Sato, T., Kamigaichi, O., and Nakamura, K.: Preseismic groundwater level changes induced by crustal deformations related to earthquake swarms off the east coast of Izu Peninsula, Japan, Geophys. Res. Lett., 31, 2004.

Kopytenko, Y. A., Matiashvili, T. G., Voronov, P. M., Kopytenko, E. A., and Molchanov, O. A.: Detection of ultra-low-frequency emissions connected with the Spitak earthquake and its aftershock activity, based on geomagnetic pulsations data at Dusheti and Vardzia observatories, Phys. Earth Planet. Inter., 77, 85–95, 1993.

Krzyszczak, J., Baranowski, P., Zubik, M., Kazandjiev, V., Georgieva, V., Cezary, S., Siwek, K., Kozyra, J., and Nieróbca, A.: Multifractal characterization and comparison of meteorological time series from two climatic zones, 1811–1824, 2019.

Lashermes, B., Jaffard, S., and Abry, P.: Wavelet leader based multifractal analysis, in: Proceedings.(ICASSP'05). IEEE International Conference on Acoustics, Speech, and Signal Processing, 2005., iv–161, 2005.

Liebovitch, L. S. and Toth, T.: A fast algorithm to determine fractal dimensions by box counting, Phys. Lett. A, 141, 386–390, 1989.

Liu, J. Y., Tsai, Y. B., Chen, S. W., Lee, C. P., Chen, Y. C., Yen, H. Y., Chang, W. Y., and Liu, C.: Giant ionospheric disturbances excited by the M9. 3 Sumatra earthquake of 26 December 2004, Geophys. Res. Lett., 33, 2006.

Lopes, R. and Betrouni, N.: Fractal and multifractal analysis: a review, Med. Image Anal., 13, 634–649, 2009.

López-Casado, C., Henares, J., Badal, J., and Peláez, J. A.: Multifractal images of the seismicity in the

Ibero-Maghrebian region (westernmost boundary between the Eurasian and African plates), Tectonophysics, 627, 82–97, https://doi.org/10.1016/j.tecto.2013.11.013, 2014.

Mandal, P., Mabawonku, A. O., and Dimri, V. P.: Self-organized fractal seismicity of reservoir triggered earthquakes in the Koyna-Warna seismic zone, Western India, Pure Appl. Geophys., 162, 73–90, https://doi.org/10.1007/s00024-004-2580-8, 2005.

Mandelbrot, B. B. and Van Ness, J. W.: Fractional Brownian motions, fractional noises and applications, SIAM Rev., 10, 422–437, 1968.

Mandelbrot, B. B.: Fractals, Form, chance Dimens., 1977.

Mandelbrot, B. B.: Multifractal measures, especially for the geophysicist, Fractals Geophys., 5–42, 1989.

Mandelbrot, B. B. and Mandelbrot, B. B.: The fractal geometry of nature, WH freeman New York, 1982.

Meng, J., Wang, C., Zhao, X., Coe, R., Li, Y., and Finn, D.: India-Asia collision was at 24 N and 50 Ma: palaeomagnetic proof from southernmost Asia, Sci. Rep., 2, 925, 2012.

Molchan, G. and Kronrod, T.: The fractal description of seismicity, Geophys. J. Int., 179, 1787–1799, https://doi.org/10.1111/j.1365-246X.2009.04380.x, 2009.

Molchanov, O. A. and Hayakawa, M.: Generation of ULF electromagnetic emissions by microfracturing, Geophys. Res. Lett., 22, 3091–3094, https://doi.org/10.1029/95GL00781, 1995.

Molchanov, O. A., Kopytenko, Y. A., Voronov, P. M., Kopytenko, E. A., Matiashvili, T. G., Fraser-Smith, A. C., and Bernardi, A.: Results of ULF magnetic field measurements near the epicenters of the Spitak (Ms= 6.9) and Loma Prieta (Ms= 7.1) earthquakes: Comparative analysis, Geophys. Res. Lett., 19, 1495–1498, 1992.

Muzy, J.-F., Bacry, E., and Arneodo, A.: The multifractal formalism revisited with wavelets, Int. J. Bifurc. Chaos, 4, 245–302, 1994.

Myint, S. W.: Fractal approaches in texture analysis and classification of remotely sensed data: Comparisons with spatial autocorrelation techniques and simple descriptive statistics, Int. J. Remote Sens., 24, 1925–1947, 2003.

Ouzounov, D., Liu, D., Chunli, K., Cervone, G., Kafatos, M., and Taylor, P.: Outgoing long wave radiation variability from IR satellite data prior to major earthquakes, Tectonophysics, 431, 211–220, 2007.

Panda, M. N., Mosher, C., and Chopra, A. K.: Application of wavelet transforms to reservoir data analysis and scaling, in: SPE Annual Technical Conference and Exhibition, 1996.

Panda, S. K., Choudhury, S., Saraf, A. K., and Das, J. D.: MODIS land surface temperature data detects thermal anomaly preceding 8 October 2005 Kashmir earthquake, Int. J. Remote Sens., 28, 4587–4596, 2007.

Pastén, D. and Pavez-Orrego, C.: Multifractal time evolution for intraplate earthquakes recorded in southern Norway during 1980–2021, Chaos, Solitons & Fractals, 167, 113000, 2023.

Pentland, A. P.: Fractal-based description of natural scenes, IEEE Trans. Pattern Anal. Mach. Intell., 661–674, 1984.

Prajapati, R.,Arora, A.: Investigation of geomagnetic field variations in search of seismo-electromagnetic emissions associated with earthquakes in subduction zone of Andaman-Nicobar, India, 2023.

Potirakis, S. M., Hayakawa, M., and Schekotov, A.: Fractal analysis of the ground-recorded ULF magnetic fields prior to the 11 March 2011 Tohoku earthquake (M W = 9): discriminating possible earthquake precursors from space-sourced disturbances, Nat. Hazards, 85, 59–86, https://doi.org/10.1007/s11069-016-2558-8, 2017.

Qiuming, C.: Fractal density and singularity analysis of heat flow over ocean ridges, Sci. Rep., 6, 1–10, https://doi.org/10.1038/srep19167, 2016.

Rahimi-Majd, M., Shirzad, T., and Najafi, M. N.: A self-organized critical model and multifractal analysis for earthquakes in Central Alborz, Iran, Sci. Rep., 12, 8364, 2022.

Rawat, G., Chauhan, V., and Dhamodharan, S.: Fractal dimension variability in ULF magnetic field with reference to local earthquakes at MPGO, Ghuttu, Geomatics, Nat. Hazards Risk, 7, 1937–1947, https://doi.org/10.1080/19475705.2015.1137242, 2016.

Rossi, G.: Fractal analysis as a tool to detect seismic cycle phases, in: Fractals and Dynamic Systems in Geoscience, Springer, 169–179, 1994.

Roy, P. N. S. and Mondal, S. K.: Multifractal analysis of earthquakes in Kumaun Himalaya and its surrounding region, Journal of earth system science., 121,1033-1047, 2012.

Rikitake, T.: Earthquake precursors, Bull. Seismol. Soc. Am., 65, 1133–1162, 1975.

Schaefer, D. W.: Fractal models and the structure of materials, MRS Bull., 13, 22–27, 1988.

Scholz, C. H., Sykes, L. R., and Aggarwal, Y. P.: Earthquake Prediction: A Physical Basis: Rock dilatancy and water diffusion may explain a large class of phenomena precursory to earthquakes., Science (80-. )., 181, 803–810, 1973.

Serrano, E. and Figliola, A.: Wavelet Leaders: A new method to estimate the multifractal singularity spectra, Phys. A Stat. Mech. its Appl., 388, 2793–2805, https://doi.org/10.1016/j.physa.2009.03.043, 2009.

Sethumadhav, M. S., Gunnell, Y., Ahmed, M. M., and Chinnaiah: Late Archean manganese mineralization

and younger supergene manganese ores in the Anmod-Bisgod region, Western Dharwar Craton, southern India: Geological characterization, palaeoenvironmental history, and geomorphological setting, Ore Geol. Rev., 38, 70–89, https://doi.org/10.1016/j.oregeorev.2010.06.001, 2010.

Shen, Y. and Tian, B.: Bilinear auto-Bäcklund transformations and soliton solutions of a (3+1)-dimensional generalized nonlinear evolution equation for the shallow water waves, Appl. Math. Lett., 122, 107301, 2021.

Smirnova, N., Hayakawa, M., and Gotoh, K.: Precursory behavior of fractal characteristics of the ULF electromagnetic fields in seismic active zones before strong earthquakes, Phys. Chem. Earth, Parts A/B/C, 29, 445–451, 2004.

Smirnova, N. A., Kiyashchenko, D. A., Troyan, V. N., and Hayakawa, M.: Multifractal Approach to Study the Earthquake Precursory Signatures Using the Ground-Based Observations, Review of Applied Physics, Hayakawa and Ida, 2013.

Sridharan, M. and Ramasamy, A. M. S.: Fractal analysis for geomagnetic secular variations, J. Indian Geophys. Union, 10, 175–185, 2006.

Stanica, D. A. and Stănică, D.: ULF pre-seismic geomagnetic anomalous signal related to Mw8.1 offshore chiapas earthquake, Mexico on 8 September 2017, Entropy, 21, https://doi.org/10.3390/e21010029, 2019.

Szczepaniak, A. and Macek, W. M.: Asymmetric multifractal model for solar wind intermittent turbulence, Nonlinear Process. Geophys., 15, 615–620, 2008.

Telesca, L., Colangelo, G., Lapenna, V., and Macchiato, M.: Monofractal and multifractal characterization of geoelectrical signals measured in southern Italy, Chaos, Solitons and Fractals, 18, 385–399, https://doi.org/10.1016/S0960-0779(02)00655-0, 2003.

Telesca, L., Lapenna, V., Vallianatos, F., Makris, J., and Saltas, V.: Multifractal features in short-term time dynamics of ULF geomagnetic field measured in Crete, Greece, Chaos, Solitons and Fractals, 21, 273–282, https://doi.org/10.1016/j.chaos.2003.10.020, 2004.

Telesca, L., Lapenna, V., and Macchiato, M.: Multifractal fluctuations in seismic interspike series, Phys. A Stat. Mech. its Appl., 354, 629–640, 2005.

Turcotte, D. L.: Fractals in geology and geophysics, Pure Appl. Geophys., 131, 171–196, 1989.

Turcotte, D. L.: Fractals and chaos in geology and geophysics, Cambridge university press, 1997.

Uyeda, S., Hayakawa, M., Nagao, T., Molchanov, O., Hattori, K., Orihara, Y., Gotoh, K., Akinaga, Y., and Tanaka, H.: Electric and magnetic phenomena observed before the volcano-seismic activity in 2000 in the

Izu Island Region, Japan, Proc. Natl. Acad. Sci., 99, 7352–7355, 2002.

Virk, H. S., Walia, V., and Kumar, N.: Helium/radon precursory anomalies of Chamoli earthquake, Garhwal Himalaya, India, J. Geodyn., 31, 201–210, 2001.

Wang, W., Cheng, Q., Tang, J., Pubuciren, Song, Y., Li, Y., and Liu, Z.: Fractal/multifractal analysis in support of mineral exploration in the Duolong mineral district, Tibet, China, Geochemistry Explor. Environ. Anal., 17, 261–276, 2017.

Wendt, H.: Contributions of Wavelet Leaders and Bootstrap to Multifractal Analysis : Images , Estimation Performance , Dependence Structure and Vanishing Moments . Confidence Intervals and Hypothesis Tests, 1–292, 2008.

Werner, D. H., Haupt, R. L., and Werner, P. L.: Fractal antenna engineering: The theory and design of fractal antenna arrays, IEEE Antennas Propag. Mag., 41, 37–58, 1999.

Weszka, J. S., Dyer, C. R., and Rosenfeld, A.: A comparative study of texture measures for terrain classification, IEEE Trans. Syst. Man. Cybern., 269–285, 1976.

Xu, T., Moore, I. D., and Gallant, J. C.: Fractals, fractal dimensions and landscapes—a review, Geomorphology, 8, 245–262, 1993.

Xu, G., Han, P., Huang, Q., Hattori, K., Febriani, F., and Yamaguchi, H.: Anomalous behaviors of geomagnetic diurnal variations prior to the 2011 off the Pacific coast of Tohoku earthquake (Mw9.0), J. Asian Earth Sci., 77, 59–65, https://doi.org/10.1016/j.jseaes.2013.08.011, 2013.

Yang, H., Pan, H., Wu, A., Luo, M., Konaté, A. A., and Meng, Q.: Application of well logs integration and wavelet transform to improve fracture zones detection in metamorphic rocks, J. Pet. Sci. Eng., 157, 716–723, https://doi.org/10.1016/j.petrol.2017.07.057, 2017.

Yen, H.-Y., Chen, C.-H., Yeh, Y.-H., Liu, J.-Y., Lin, C.-R., and Tsai, Y.-B.: Geomagnetic fluctuations during the 1999 Chi-Chi earthquake in Taiwan, Earth Planets Space, 39–45 pp., 2004.