# Peer review of "Fractal analysis of geomagnetic data to decipher pre-earthquake process in 1 Andaman-Nicobar region, India 2 Rahul Prajapati1\* and Kusumita Arora1 3 1 Geomagnetism Group, CSIR-National Geophysical Research Institute, Hyderabad-500007, India; 4 rahulphy007@gmail.c"

_Nonlinear Processes in Geophysics, 2024_

## Referee Comment (RC1)

**Comments on 'Fractal analysis of geomagnetic data to decipher pre-earthquake process in Andaman-Nicobar region, India' (npg-2024-8) authored by Rahul Prajapati and Kusumita Arora.**

From the measures of fractal and multifractal dimensions of observed Z-component seismo-electromagnetic (EM) signatures prior to earthquakes, the authors tried to study the possible existence of seismic precursor. Although their study is fine and interesting, the manuscript cannot be accepted for publication just in the present form due to the following several major problems which made me unable to follow their studies well. Hence, I cannot judge if the study is acceptable or not. The authors should substantially re-write and re-organize the manuscript and then re-submit it.

**Major Problems**
(1) The authors applied two methods to measure the fractal dimensions. They should simply describe the methods and clearly explain the parameters. For example, the authors should explain the definitions of 'length' and 'k' in Figure 1.
(2)The authors must use a testing example to describe the way applied to estimate the values of multifractal spectrum, i.e., $h_w$, and to explain whether or not the estimated values are reliable. This will help me to accept the results.
(3)The English writing should be substantially re-written because there are many grammatical and typo errors. Meanwhile, the statements should be re-organized
(4)In Table 1, the authors should replace 'Mod' and 'Large' for Mag (magnitude), 'Mod', 'Shallow', and 'Large' for 'Foc. D.' (Focal Depth),' and 'Mod', 'Small', ad 'Large' for 'Epi. D.' (Epicentral Distance)' by the magnitude range, focal depth range, and epicentral range in numbers.

**Minor Problems**
(1)The abstract is not concise.
(2)It is better to provide a figure to show an example of observed Z-component seismo-electromagnetic (EM) signatures.
(3)The quality of figures should be improved.

---

## Author Comment (AC1)

Dear Editor-in-Chief,

We take this opportunity to thank you, and Referee 1 for thoughtful comments on our manuscript which helped us in improving the manuscript. We hope that the answer of each major and minor comment will meet your expectations. The comments of the reviewers and their replies are listed here one by one, which includes some figure also.

Yours sincerely,

Rahul Prajapati, Kusumita Arora

**Major Problems:**

**Comment 1**. The authors applied two methods to measure the fractal dimensions. They should simply describe the methods and clearly explain the parameters. For example, the authors should explain the definitions of 'length' and 'k' in Figure 1.

**Answer 1.** We have revised the methodological section and incorporated the sentences and equations which describe the methods and clearly explain the parameters involved in both the methods. The revisions also include the definition of 'length' and 'k' used in Figure 1. The revised section of methodology is attached at the end of all comments and answers (Page 10-14). The methodological section of manuscript has also revised accordingly.

**Comment 2**. The authors must use a testing example to describe the way applied to estimate the values of multifractal spectrum, i.e., $h_w$, and to explain whether or not the estimated values are reliable. This will help me to accept the results.

**Answer 2:** For the testing example, we have taken the 128 data samples of vertical component of geomagnetic field on 13 May, 2019 and 01:00:00 to 01:02:08 hrs. (Figure 1 f) to explain the way multifractal spectrum values (hw) is estimated. The estimation of multifractal spectrum using wavelet leader technique comprises of following four steps:

(i)    In the first step we applied the discrete wavelet transform and decomposed the signal at five levels and restored the values of detail and approximation wavelet coefficients (Figure 1 a-f).

[Figure]

**Figure 1.** the test signal (f) and its decomposition at level 1 to 5 (e to a) using DWT transform.

(ii)    The detail wavelet coefficient is used for computation of wavelet leaders ($w_l$) from each scale shown in Figure 2.

[Figure]

**Figure 2**. Wavelet leader selected from detail wavelet coefficients at level 1 to 5 (from top).

(iii) The $w_l$ estimated at each scale is used to compute multiresolution structure function of multifractal parameter $\varphi_q, D_q, H_q,$ and $C_p$ at linearly space moment order (q=-5 to +5), in which $D_q, and\ H_q$ are the parameters of the multifractal spectrum. The equations involved to compute these parameters are explained clearly by Jaffard et al. (2007) and Serano and Figliola (2009). The variation of $D_q, and\ H_q$ from scale 2 to 5 at moment order q is shown in Figure 3a and b respectively.

[Figure]

**Figure 3**. The variation in multifractal parameter (a) $D_q$ and (b) $H_q$ with moment order q at level 2 to 5.

(iv)    At this stage, we have the values of multifractal parameters at scale one to five and moment order q. The final values of multifractal parameters correspond to q (-5 to +5) is the slope of linear regression of multifractal parameters measured at different scales verses log of scales. Thus, each value of multifractal parameters ($\varphi_q, D_q, H_q,$ and $C_p$) are now available with respect to moment order q(-5 to +5). The variation of $\varphi_q, D_q, and\ H_q,$ with respect to q is shown in Figure 4 a-c respectively, and multifractal spectrum ($H_q,$ vs $D_q$) shown in Figure 4d.

[Figure]

**Figure 4.** The variation in final multifractal parameters (a) $\varphi$ , (b) $D$, (c) $H$ with respect to moment order q and the spectrum of multifarctal parameter ($D$ $vs. h$ ) is shown in (d).

To further establish the reliability of the computed multifractal spectrum values, we have tested this method on four different types of synthetic signals with known scaling exponents h1(0.2), h2(0.4), h3(0.6), and h4 (addition of h1, h2, and h3 in series). The small exponent indicates the less correlated or noisier signal, whereas signal of large exponent indicates high correlated or more smooth (Figure 5) data. For multifractal, the disturbed signals are expressed through higher degree of multifractal nature or large spectrum width than the spectrum width of less disturb or smooth signal i.e. spectrum width of h4>h1>h2>h3. Thus, we can say that the values are reliable and can fulfil the objective on application of geomagnetic data.

[Figure]

**Figure 5**. The synthetic signal generated at hurst exponent (a) 0.2, (b) 0.4, (c) 0.5, and (d) combination of all three in series.

[Figure]

**Figure 6.** The multfarctal spctrum of signal h1, h2, h3, and h4 showing the degree of multifractality**.**

**Comment 3.** The English writing should be substantially re-written because there are many grammatical and typo errors. Meanwhile, the statements should be re-organized

**Answer 3.** We have improved English syntax throughout the manuscript.

**Comment 4.** In Table 1, the authors should replace 'Mod' and 'Large' for Mag (magnitude), 'Mod', 'Shallow', and 'Large' for 'Foc. D.' (Focal Depth),' and 'Mod', 'Small', ad 'Large' for 'Epi. D.' (Epicentral Distance)' by the magnitude range, focal depth range, and epicentral range in numbers.

**Answer 4.** Table 1 is revised and also the ranges of magnitude, focal depth, and epicentral distances listed in table caption. The revised table is incorporated at the end of this comment and answer section (Page 15).

**Minor Problems**

**Comment 5.** The abstract is not concise.

**Answer 5.** We have re-written the abstract. The revised abstract is reduced to 187 words from 202 words of original abstract as per norm of journal (100-200 words).

The revised abstract as follows:

"The emission of seismo-electromagnetic (EM) signatures prior to earthquake recorded in geomagnetic data has potential to reveal the pre-earthquake processes. This study focused to analysis of vertical component of a geomagnetic field from Mar 2019 to Apr 2020 using fractal and multifractal approach to identify the EM signatures in Campbell Bay, a seismically active region of Andaman and Nicobar. The significant enhancements in monofractal dimension and spectrum width components of multifractal highlights the complex nature of geomagnetic field due to interference of high frequency EM field, due to pre-earthquakes processes of micro fracturing of the shallow crust in the vicinity of the West Andaman Fault and Andaman Trench. On the other hand, the enhancements in holder exponents, highlight the complexities in the geomagnetic time series due to interference of less correlated, smooth, and low frequency EM field, suggesting that pre-earthquake processes on Seulimeum Strand (SS) are dominated by electrokinetic processes. The mono fractal, spectrum width, and holder exponent parameter reveals different nature of pre-earthquakes process prior to earthquakes with an average of 10, 12, and 20 days respectively, which are also lies in range of short -term earthquake prediction."

**Comment 6.** It is better to provide a figure to show an example of observed Z-component seismo-electromagnetic (EM) signatures.

**Answer 6.** To observe the EM signatures in vertical component of geomagnetic field in night time data (22:00-02:00), we have selected two quite days (25 May and 3 Aug, 2019) in which one ($25^{th}$ May) is interfered by EM field, while second (3 Aug) is not interfered by EM field. Figure 7a, b showing the field on and clearly deciphers the significant fluctuations in the field on $25^{th}$ May, 2019 even on night time quite data, while field on $3^{rd}$ Aug, 2019 does not showing such fluctuations on quite day. A significant enhancement in hw (Figure 7c) and hwp (Figure 7d) also marked on $25^{th}$ May, 2019, while there in no such enhancements marked in hw and hwp on on $3^{rd}$ Aug, 2019. This example of observation will be also included in manuscript.

[Figure]

**Figure 7.** The night time data of vertical component of geomagnetic field on (a) 25th May, 2019 and (b) 3rd Aug, 2019. The multifractal component of (a) hw, (b) hwp, and (c) hwn from Mar, 2019 to April, 2020.

**Comment 7.** The quality of figures should be improved

**Answer 7.** All Figures in manuscript are 300 dpi. The resolution of Figure in manuscript will be enhanced by 600 dpi at the time of submission of revised manuscript.

**Revised part methodology section:**

**2. Methodological Approach**

It is proposed to apply both fractal and multifractal approaches to the Z component time series, to distinguish between the different source characteristics and examine their relationship to earthquake parameters. The Z-component of 1 Hz geomagnetic signal is preferred because it is more prone to be affected by the local EM field generated by lithospheric deformation.

Gotoh et al. (2003) tested different methods for estimation of fractal dimension of geomagnetic signal and suggested that the fractal dimension value using Higuchi method, is more reliable and consistent than others. In Higuchi method, a time series x(n) is decomposed into time series of different lengths $x_k^m$, defined as:

$$x_k^m: x(m), x(m+k), x(m+2k), \dots \dots x(m + \left(\frac{N-k}{k}\right).k),$$

Where, n is 1,2 ,3 …N, $m$ is 1,2,3…$k$, and $k$ is 1,…., $k_{max}$. The average length of decomposed time series $L_m(k)$ computed at interval of time from $k = 1$ to $k_{max}$ are related to each other as:

$$L(k) \propto k^{-f_D}$$

Where L is average length of decomposed time series , $f_D$ is fractal dimension equal to the slope of fitted line over $\log(L(k))$ versus $\log\left(\frac{1}{k}\right)$.

In our analysis, we have adopted the Higuchi method for monofractal analysis. Application of Higuchi method on night-time (22:00-02:00 LT) Z-component of geomagnetic signal of 3 April 2019, is shown in Figure 2 to verify the appropriateness of this approach.

[Figure]

**Figure 2.** The linear fitting over log of average length and log of size of time interval (scale) showing the power law nature of geomagnetic signal.

Muzy et al. (1994) proposed an approach for multifractal analysis based on discrete wavelet or wavelet leader. In this approach, the local suprema $f_{i,k}$ is obtained from discrete wavelet coefficients at dyadic scales, where, $k$ is translation parameter, $i$ is scale, and the position in time for dyadic interval is $2^i k$ (Jaffard et al., 2006; Wendt et al., 2008). The local suprema of wavelet coefficients $f_{i,k}$ obtained at dyadic scale, aid in computation of the multiresolution structure function $S_{xL}(q,i)$ for to produce the global holder exponent (Serrano and Figliola, 2009) i.e.

$$S(q,i) \sim (2^i)^{\tau(q)}$$

Where, i is scale, q is moment and $\tau(q)$ is scaling exponent. The scaling exponent follows power law relation can be estimated by following relation

$$\tau(q) = \lim_{i \to 0} \inf \frac{log(S_{xL}(q,i))}{log(2^i)}$$

The spectrum of global holder exponent is derived from multifractal formalism using legendre function (Serrano and Figliola, 2009), which leads to.

$$f(\alpha) = \inf(1 - \tau(q) + \alpha(q) * q),$$

Where $\alpha$ is global holder exponent and $f(\alpha)$ is function of global holder exponent. The degree of intermittency or multifractality is defined by multifractal or singularity spectrum i.e. $\Delta \alpha = \alpha_{max} - \alpha_{min}$. Larger the width of multifractal spectrum, larger is the multifractality or intermittency, and vice-versa. The width of multifractal spectrum $h_w$ (from $-q\ to + q$) indicates the overall degree of multifractality of signal. The spectrum width $h_{wp}$ ( $q > 0$) and $h_{wn}$ ( $q < 0$) indicates the weaker and stronger singularity of multifractal signal. The $h_{max}$-$h_{min}$ curve defines the average fluctuations embedded in the signal while $h(0)$ represents the zero-order exponent or monofractal dimension (Hayakawa et al., 1999). Similarly, $f_{max}$ define the exponent which occurred maximum number of times. Application of multifractal on nighttime (22:00-02:00 LT) Z-component of geomagnetic signal of 3 April 2019, is shown in Figure 3. Thus the wavelet leader approach is adopted in this study due to contact support for wide range of $q$ ($-q\ to + q$) and stability for scaling function for negative $q$ values compared to other techniques.

[Figure]

**Figure 3.** The multifractal analysis for 1800 samples of 3$^{rd}$ April 2019, where (a) The variation of holder exponent (h) with moment order q in range of -15 to +15 showing as $h_{min}$, $h_{max}$, and $h(0)$. (b) Multifractal spectrum showing the width of spectrum $h_w$, $h_{wp}$ and $h_{wn}$.

From the same data, from fractal analysis, the power law behaviour, and from multifractal, the finite width of multifractal spectrum and variation in holder exponent demonstrates the fractal as well as multifractal natures of the signal.

The fractal dimension $f_D$ of the total duration of Z-component data is calculated for consecutive time windows of 30 min to trace the variations of the fractal dimension, producing eight values for each day. The choice of a 30 min time window (consisting of 1800 data points) is based on the balance between the stability of fluctuations in fractal dimension and minimizing loss of information after trials with 15 min and 1 hr. time windows.

Similarly, the spectrum width parameter ($h_w$, $h_{wp}$, $and$ $h_{wn}$) and holder exponent parameter $h_{max}$, $h_{min}$ and, $h(0)$ estimated for the total length of Z component from window of 30 minute to identify the degree of singularity or complexity (global, weaker, and stronger) as well as degree of fluctuations with respect to amplitude (from smaller to larger). The shorter fluctuations in fractal dimensions are smoothed by applying a 15-day moving mean.

The increments in fractal dimension and multifractal parameter (spectrum width and holder exponent) value greater than the threshold value ($\mu + \sigma$) are considered as a significant evidence of existence of EM signals of lithospheric origin.

**Table 1:** The following table summarizes the earthquake and its characteristics presence (Y) or absence (-) of potential enhancements in monofractal ($f_D$) and multifractal ($h_w, f_{max}, h_{max}$) components and diurnal ratio. The characteristics of earthquakes are given by it range of Magnitude (moderate: 4.5≤M<5, large: M≥5.0), focal depth (shallow: f≤25km, moderate: 25≤f<80km, large: f≥80km), and epicentral distance (small: ed≤60km, moderate: 60<ed≤150, large: ed>150).

| EQ. No. | Magnitude | Focal Depth (Km) | Epicentral Distance (Km) | Single/ Cluster | $f_D$ | $h_w$ | $f_{max}$ | $h_{max}$ | Diurnal ratio |
|---|---|---|---|---|---|---|---|---|---|
| 1-45 | Moderate to Large | Moderate | Moderate | C | - | Y | Co- | - | - |
| 46-48 | Moderate | Moderate | Moderate | C | - | - | Y | Y | Y |
| 49 | Moderate | Moderate | Moderate | S | Co- | Y | - | - | Y |
| 50-51 | Moderate | large/ shallow | Large | C | Y | Y | - | - | Post- |
| 52 | Moderate | Shallow | Large | S | - | - | - | - | Y |
| 53-54-55 | Moderate to Large | Shallow | Small to Moderate | C | Y | Y | Y | Y | Y |
| 56 | Moderate | Moderate | Large | S | Y | - | - | - | - |
| 57 | Large | Shallow | Large | S | Y | - | - | - | - |
| 58 | Large | Large | Moderate | S | Y | - | - | - | - |
| 59 | Moderate | Shallow | Large | S | - | - | - | - | Y |
| 60 | Moderate | Large | Moderate | S | Co- | - | - | - | Y |
| 61 | Moderate | Shallow | Large | S | Co- | - | - | - | Y |
| 62 | Moderate | Shallow | Large | S | - | - | - | - | - |
| 63 | Moderate | Shallow | Large | S | - | - | - | - | post |

---

## Author Response (AR1)

**Referee 1:**

Dear Editor-in-Chief,

We take this opportunity to thank you, and Referee 1 for your thoughtful comments on our manuscript which helped us in improving the manuscript. We hope that the answer of each major and minor comment will meet your expectations. The comments of the reviewers and their replies are listed here one by one. The line number mentioned in reply of each comment is correspond to revised plain manuscript and it may vary from line number of track change revised manuscript. The comments of referee are in red color text and answer of author is in normal font and black color.

**Major Problems:**

**Comment 1**. The authors applied two methods to measure the fractal dimensions. They should simply describe the methods and clearly explain the parameters. For example, the authors should explain the definitions of 'length' and 'k' in Figure 1.

**Answer 1.** We have revised the methodological section and incorporated the sentences and equations which describe the methods more clarity and explain the parameters involved in both the methods. The revisions also include the definition of 'length' and 'k' used in Figure 1. The revised methodological section is incorporated in the revised manuscript accordingly (line number 128-175).

**Comment 2**. The authors must use a testing example to describe the way applied to estimate the values of multifractal spectrum, i.e., $h_w$, and to explain whether or not the estimated values are reliable. This will help me to accept the results.

**Answer 2:** For the testing example, we have taken the 128 data samples of vertical component of geomagnetic field on 13 May, 2019 and 01:00:00 to 01:02:08 hrs (figure 1 f) to explain the way multifractal spectrum values (hw) is estimated. The estimation of multifractal spectrum using wavelet leader technique comprises of following four steps:

(i)      In the first step we applied the discrete wavelet transform and decomposed the signal at five levels and restored the values of detail and approximation wavelet coefficients (Figure 1 a-f).

[Figure]

**Figure 1.** the test signal (f) and its decomposition at level 1 to 5 (e to a) using DWT transform.

(ii) The detail wavelet coefficient is used for computation of wavelet leaders ($w_l$) from each scale shown in figure 2.

[Figure]

**Figure 2**. Wavelet leader selected from detail wavelet coefficients at level 1 to 5 (from top).

(iii) The $w_l$ estimated at each scale is used to compute multiresolution structure function of multifractal parameter $\varphi_q, D_q, H_q,$ and $C_p$ at linearly space moment order (q=-5 to +5), in which $D_q, and\ H_q$ are the parameters of the multifractal spectrum. The equations involved to compute these parameters are explained clearly by Jaffard et al. (2007) and Serano and Figliola (2009).

(iv) The variation of $D_q$, $and\ H_q$ from scale 2 to 5 at moment order q is shown in figure 3a and b respectively.

[Figure]

**Figure 3**. The variation in multifractal parameter (a) $D_q$ and (b) $H_q$ with moment order q at level 2 to 5.

(v) At this stage, we have the values of multifractal parameters at scale one to five and moment order q. The final values of multifractal parameters correspond to q (-5 to +5) is the slope of linear regression of multifractal parameters measured at different scales verses log of scales. Thus, each value of multifractal parameters ($\varphi_q, D_q, H_q,$ and $C_p$) are now available with respect to moment order q(-5 to +5). The variation of $\varphi_q, D_q, and\ H_q,$ with respect to q is shown in figure 4 a-c respectively, and multifractal spectrum ($H_q,$ vs $D_q$) shown in figure 4d.

[Figure]

**Figure 4.** The variation in final multifractal parameters (a) $\varphi$ , (b) $D$, (c) $H$ with respect to moment order q and the spectrum of multifarctal parameter ($D\ vs.\ h$ ) is shown in (d).

To further establish the reliability test of the computed multifractal spectrum values, we have tested this method on four different types of synthetic signals with known scaling exponents h1(0.2), h2(0.4), h3(0.6), and h4 (addition of h1, h2, and h3 in series). The small exponent indicates the less correlated or noisier signal, whereas signal of large exponent indicates high correlated or more smooth (figure 5) data. For multifractal, the disturbed signals are expressed through higher degree of multifractal nature or large spectrum width than the spectrum width of less disturb or smooth signal i.e. spectrum width of h4>h1>h2>h3. Thus, we can say that the values are reliable and can fulfil the objective on application of geomagnetic data.

[Figure]

**Figure 5**. The synthetic signal generated at hurst exponent (a) 0.2, (b) 0.4, (c) 0.5, and (d) combination of all three in series.

[Figure]

**Figure 6.** The multfractal spctrum of signal h1, h2, h3, and h4 showing the degree of multifractality.

A numerical simulation (synthetic test) of fractal and multifractal on fBm signals performed and has been incorporated to revised manuscript (line 225-233 and line 268-272) and supporting document.

**Comment 3.** The English writing should be substantially re-written because there are many grammatical and typo errors. Meanwhile, the statements should be re-organized

**Answer 3.** We have improved English syntax throughout the manuscript.

**Comment 4.** In Table 1, the authors should replace 'Mod' and 'Large' for Mag (magnitude), 'Mod', 'Shallow', and 'Large' for 'Foc. D.' (Focal Depth),' and 'Mod', 'Small', ad 'Large' for 'Epi. D.' (Epicentral Distance)' by the magnitude range, focal depth range, and epicentral range in numbers.

**Answer 4.** Table 1 is revised and also the ranges of magnitude, focal depth, and epicentral distances listed in table caption. The revised table is incorporated at the end of this comment and answer section (Page 29).

 **Minor Problems**

**Comment 5.** The abstract is not concise.

**Answer 5.** We have re-written the abstract. The revised abstract is now included with 200 words, which also falls under the journal's norm (100-200 words). The modified abstract is included in revised manuscript (line 9-22).

**Comment 6.** It is better to provide a figure to show an example of observed Z-component seismo-electromagnetic (EM) signatures.

**Answer 6.** To observe the EM signatures in vertical component of geomagnetic field in night time data (22:00-02:00), we have selected two quite days (25 May and 3 Aug, 2019) in which one (25th May) is interfered by EM field, while second (3 Aug) is not interfered by EM field. Figure 7a, b showing the field on and clearly deciphers the significant fluctuations in the field on 25th May, 2019 even on night time quite data, while field on 3rd Aug, 2019 does not showing such fluctuations on quite day.  A significant enhancement in hw (figure 7c) and hwp (figure 7d) also marked on 25th May, 2019, while there in no such enhancements marked in hw and hwp on on 3rd Aug, 2019. This example of observation has been also incorporated in revised manuscript (line 410-415) and supporting document.

[Figure]

**Figure 7.** The night time data of vertical component of geomagnetic field on (a) 25th May, 2019 and (b) 3rd Aug, 2019. The multifractal component of (a) hw, (b) hwp, and (c) hwn from Mar, 2019 to April, 2020.

**Comment 7.** The quality of figures should be improved.

**Answer 7.** We have replaced all the figures in the revised manuscript with modified high-resolution figures.

**Referee 2:**

Dear Editor-in-Chief,

We take this opportunity to thank you, and Referee 2 for your thoughtful comments on our manuscript which helped us in improving the manuscript. We hope that the answer of each major and minor comment will meet your expectations. The comments of the reviewers and their replies are listed here one by one. The line number mentioned in reply of each comment is correspond to revised plain manuscript and it may vary from line number of track change revised manuscript. The comments of referee#2 are in red color text and answer of author is in normal font and black color.

Yours sincerely,

Rahul Prajapati, Kusumita Arora

**Referee#2. Comment and Answer:**

I have checked the present work. The topic addressed is well-known in literature and of particular importance. A series of flaws arise that I would like to ask authors to consider them in their revision. These are listed below:

**Comment 1-** The importance of fractals must be well-introduced, justified and elaborated. It is applied widely in several fields including seismology and earthquakes sciences. Applications of fractal geometry and fractal dimensions to study various seismic activities have been also explored in details in various studies based on dissimilar methodologies See the present missed references in the field.

Chaos, Solitons & Fractals **14**: 917-928 (2022); Acta Mech. **233**:2107-2122 (2022); Geophys. J. Int. **179**(3): 1787-1799 (2009); Phil. Trans.: Phys. Sci. Eng. **348**(1688): 449-457 (1994); Chaos, Solitons & Fractals **167**: 113000 (2023); Chaos **31**: 043124 (2021); Nat. Haz. Earth Syst. Sci. **23**: 1911-1920 (2023)

Multifractal measures, especially for geophysicist. In: Scholz, CH, Mandelbrot BB (eds) Fractals in geology and geophysics, Birkhäuser Verlag, Basel, pp. 5-42.

Please justify the importance of fractals and multifractals in sciences. See the missing references

The Fractal Dimensionality of Seismic Wave. In: Yuan C, Cui J and Mang HA (eds). Computational Structural Engineering, Springer, Dordrecht.

Fractal models of earthquakes dynamics. Review of Nonlinear Dynamics and Complexity (eds) Schuster HG, pp. 107-158, Wiley-VCH Verlag GmbH & Co. KGaA, Weinheim.

A fractal model of earthquake occurrence: Theory, simulations and comparison with the aftershock data. J. Phys.: Conf. Ser. **319**, 012004.

Fractal Concepts and their Application to Earthquakes in Austria. In: Lehner, F.K., Urai, J.L. (eds) Aspects of Tectonic Faulting. Springer, Berlin, Heidelberg, 2000

Fractal concepts in surface growth. Cambridge University Press., 1995.

Scienze Fisiche Naturali. 31(1):203–9. (2020); Cont. Mech. Therm **34**: 1219-1235 (2022); . Sci. Rep. **10**: 21892 (2020); Remote Sens. **11**, 2112 (2019); Dynamics of Atmospheres and Oceans 106, 101459 (2024); Tectonophysics. 722:154–62 (2017); Pure Appl Geophysics. 172(7):1909–21 (2015); *Pure Appl. Geophys.* **176**, 2739–2750 (2019); Hydrobiologia 851, 2543–2559 (2024); Chaos Solitons and Fractals 178, 114317 (2024); Thermal Science and Engineering Progress 45, 102145 (2024)

**Answer 1.** We appreciate the refree#2 for this suggestion to include the study of application of fractals and multifractals in field of seismology. The various application of fractals in science as well as infield of earthquake and seismology introduced in the revised version of manuscript (line 76-96). The references also incorporated accordingly.

 **Comment 2.** The methodological schemes addressed in Section 2 requires a careful rewritten. It is not really clear what authors aim to.

**Answer 2.** We have revised the methodological section and incorporated all relevant sentences and equations to describe the methods with more clarity. All the revised of methodology in incorporated in revised manuscript (line number 128-175).

**Comment 3.** The analysis done is fine, however, can we improve the numerical simulations? Can we dress a table clarifying data used?

**Answer 3.** For the numerical simulation of fractal and multifractal analysis in present study, we preferred to simulate four different types of monofractal signals with known scaling exponent h1(0.2), h2(0.4), h3(0.6), and a multifractal signal h4 (addition of h1, h2, and h3 in series). The small exponent indicates the less correlated signal or noisier than signal of large exponent indicates high correlated or smoother (Figure 1). From the theoretical approach, the fractal dimension of more noiser or less correlated signal should be larger than smoother or correlated signal. The fractal dimension of h1, h2, and h3 calculated from Higuchi method is 1.7, 1.6, and 1.4, while for h4 is 1.6 (Figure 2). For multifractal signal h4 the fractal dimension is lower than the h3 even it is more heterogeneous than h3. From the concept of multifractal, the more noisy or heterogeneous signal encompasses through higher degree of multifractal nature and large spectrum width than the spectrum width of less disturb or smooth signal i.e. spectrum width of h4>h1>h2>h3. The spectrum width computed with the same procedure as discussed above is shown in Figure 3, which clearly deciphers that the spectrum width of h4>h1>h2>h3. Thus, the multifractal analysis shows the true and generalised nature of heterogeneity of multifractal signal from width of spectrum. Thus, the testing of synthetic signal using fractal and multifractal approach indicates the efficacy of method to reveal the degree of complexity or heterogeneity or disturbances in signals.

[Figure]

**Figure 1**. The synthetic signal generated at hurst exponent (a) 0.2, (b) 0.4, (c) 0.5, and (d) combination of all three in series.

[Figure]

**Figure 2.** Fractal dimension of synthetic signal h1, h2, h3, and h4 from Higuchi method.

[Figure]

**Figure 3.** The multifractal spctrum of signal h1, h2, h3, and h4 showing the degree of multifractality**.**

The above discussed numerical simulation of fBm signal and its analysis with fractal and multifractal approach have been incorporated into the revised manuscript (line 225-233 and line 268-272) and supporting document. The earthquake CatLog used in the present study is added in supplementary as T1 and Table T2 – T4 summarises the correlation of enhancements in fractal dimension and each parameter of multifractal component.

**Comment 4.** Regarding Holder exponent, this is an important factor. The analyses done seem not totally clear. How it is related to fractal dimensions? any estimate for the fractal dimension anyway from observations? What about variations of the Hurst exponent?

**Answer 4.** The Holder exponent is a set of Hurst exponent i.e. the generalised version of Hurst exponent, which has efficacy to estimate the generalised nature of multifractal signal. The range of variations, maximum and minimum values of Hurst or Holder exponents, contain the information of different characteristics of the signal (discussed in methodology section). In present study we used Holder exponent to analyse all different characteristics of heterogeneity of signal.

From the spectrum method of calculation of fractal dimension, the slope obtained from log-log plot between power of signal and its frequency component is called Hurst exponent, and this Hurst exponent is directly related to fractal dimension from following relation

$$H=5-2D$$

Where H is Hurst exponent and D is fractal dimension.

In the present study, we have estimated the monofractal dimension using Higuchi method because it is more reliable than other methods for time series data (discussed in methodology section). In figure 4 we have observed variations in monofractal dimensions, where the significant enhancements are observed at seven instances. These significant enhancements in fractal dimensions indicate the nature of heterogeneity of high frequency characteristics possibly associated with micro-fracturing processes prior to earthquakes. The variation in Hurst exponents is termed as a holder exponent, which is used for delineation of different characteristics of heterogeneity embedded in the signals.

**Comment 5** - Any relation between the energy of earthquake swarm and the Hurst exponent of random variations of the magnetic field of the region studied? Earthquakes represent this change in state of equilibrium which are commonly perceived to occur due to the sudden release of energy in highly stressed zones and they repeatedly occur until the system is once again back to its equilibrium state.

**Answer 5** – We appreciate to reviewer the for this comment. To find a relation between energy of earthquake swarm and variation in Hurst exponent, we required a long duration data and the occurrences of earthquake swarms for 5-6 times in the same duration and different magnitude range of earthquakes. In the present study, we have data for duration of 14 months only and range of magnitude in only 4.5-5.3. Thus, we believe that the available data is not enough to establish the relation between earthquake energy and variation in Hurst exponent.

I would like to read the revised version of this work

---

## Author Response (AR2)

We have addressed the valuable comments of the reviewer's line by line and made corresponding modifications to the manuscript. The comments are displayed in red color, the author's response in black color.

**Comment 1.** In the introductory text, considering fractal analysis as a mathematical tool to deal with the main problem must be more elaborated. I think also that several works have been done in; literature in this direction. Some missed references are:

Geophys. J. Int. 167, 1215–1219 (2006); Pure Appl. Geophys. 162, 73–90 (2005); J. Earth Syst. Sci. 128, 22 (2019); Acta Mech 233, 2107–2122 (2022); Int. J. GeoInform. 9, 384 (2020)

**Answer 1**. We have revised the introduction section and elaborated the fractal's efficacy to deal with the problem. We have also incorporated the suggested references and few additional references also (line number 90-173). In the revised part of introduction, we have added two paragraph. In the first paragraph, we trace the evolution of fractals from their initial use in assessing natural geometries (such as clouds, coastlines, and mountain surfaces) to their diverse applications across scientific domains. These include medical science, material science, telecommunications, environmental science, computer graphics, and Earth sciences—with a focus on seismology and earthquake precursor studies. The second paragraph delves into fractal methods, starting from traditional approaches like box counting and Hausdorff method, and progressing to advanced techniques such as power spectrum analysis, Detrended Fluctuation Analysis (DFA), and the Higuchi method. We also address the limitations of these methods when applied to multifractal geometries. Finally, we briefly explore multifractal methods and their applications in geosciences.

**Comment 2.**   line 92: Fractal method aid the study of the complex nature...Which method?

**Answer 2.** We have revised the sentence which address the details about the common methods of fractal used in study of complex nature of tectonics (line number 112-118). The revised sentence highlights popular fractal methods such as Hausdorff dimension, box counting, correlation dimension,

Higuchi fractal dimension, used in study of complex nature of Earth's system. These methods help to decipher the seismicity patterns in various tectonic environments as well as before and after the main earthquake event.

**Comment 3.** What we learn from Figure 1?

**Answer 3:** Figure 1 depict the area of our study, the spatial distribution of earthquakes surrounding the geomagnetic station, geometry of major fault systems and bathymetry.

**Comment 4.** Equations must be numerated 1, 2,... After each equation, please put a dot or a comma, depending on the subsequent sentence, e.g. after equation (i), Where must be where, etc... Please check the whole manuscript. By the way, (i) is not an equation.

**Answer 4.** The equations are corrected throughout manuscript (line number 342-408).

**Comment 5.** Origin of data in Figure 2 must be mentioned by a reference.

**Answer 5.** Figure 2 is an example of geomagnetic data showing power law and self-affine nature, based on our own data.

**Comment 6.** Most of the equations addressed must be clarified and discussed. They fall from nowhere.

**Answer 6.** We have revised the second part of section 2 and elaborated all the equations used for computation of multifractal spectrum from wavelet coefficients and wavelet leaders (line number 295-376). In the revised section, we described the limitation of Holder exponent in formulation of multifractal spectrum (using the increment in spatial or time domain) due to band limited and discrete time sampled data. Further, introduced the role of structure function to resolve the issue. Again we have discussed the multifractal formulation from wavelet coefficient (using DWT) by highlighting its advantage over increment function. We again discussed the limitation of wavelet coefficients in formulation of multifractal spectrum (does not support holder exponent for $q<0$). Finally, we discussed the wavelet leader techniques to overcome the limitation of wavelet coefficients for

multifractal formulation. The multifractal formulation using wavelet leader techniques supports the holder exponent for both $q > 0$ and $q < 0$.

**Comment 7**. Fractal dimension is mentioned in line 219. However, the basic idea of fractal dimension and its properties must be addressed in the introductory text. Various definitions are found in literature. This is an important notion that deserves to be mentioned clearly. Please check also the missing references: Commun Earth Environ 5, 146 (2024); Nat. Hazards 77, 33–49 (2013); J. Asian Earth Sci. 58, 98–107 (2012); Bulletin of the Seismological Society of America 92(8):3318-3320 (2002); Fractal Fract. 8(5), 252 (2024); Earthquake Science 37, 107-121 (2024); Pure Appl. Geophys. 176, 2739–2750 (2019); J Geol Soc India 78, 226–232 (2011)

**Answer 7.** We have incorporated a paragraph in Introduction, which comprises the details of basics of fractal dimension methods such as box counting and Hausdorff. In addition, we have also incorporated the details about common fractal and multifractal methods used in analysis of geomagnetic signal. We have also incorporated the suggested references relevant to present study. (line number 153-172)

**Comment 8.** Discussion section is well-done. However, I would like to see more discussion concerning the confrontation of the fractal methodology used with observations. How fractal dimensions have been used in the present work? any estimation about their values? how they are correlated to the geometry of the region studied?

**Answer 8.** We have revised and incorporated the discussion about fractal methodology and their confrontation with observation several previous study, and also discussed about challenges and ambiguity in the interpretation of earthquake precursor studies. The previous studies in these domain supports and motivates to conduct such studies. The methodological section incorporates the steps involve in computation of fractal dimension. The changes in fractal and multifractal parameters of geomagnetic data indicates the different characteristics of SEM signatures to link the lithospheric processes in the region of earthquake preparation zones (explained briefly in discussion section).

**Comment 9.** The statistical self-affine fractal yielded from the power spectrum deserves also to be more elaborated.

**Answer 9.** In the present work, we have computed the self-affine fractal dimension of geomagnetic signal using Higuchi method, not from power spectrum method. The previous study (Hattori et al., 2004a; Gotoh et al., 2003; Smirnova et al., 2004) has also mentioned that Higuchi method is more consistent and relaible than other methods, and accordingly we have used Higuchi method in our study. Thus, we believe that the incorporation of self affine fractal dimension from power spectrum method in the manuscript is irrelevant.

**Comment 10.** Is there any estimate of the evolution of the signal energy with scale, together with the relationships between the features advocated?

**Answer 10.** In the present study, a notable and significant increase and then decrease to its original value is observed in fractal, multifractal and diurnal ratio parameters. For example, diurnal ratio anomaly increase from 1.21 to 1.26 then reached to 1.21 to 1.31 then return to original value 1.21 prior to earthquake 52 and 61. Similarly, enhancements in $f_D$ and $h_w$ observed from 1.35 and 0.24 and reached to 1.38 and 0.25 then return to its original value prior to earthquake 56 and 49 respectively. A similar observation also marked prior to the clustered of earthquake 46-48 and 53-55. These increments are attributed to the evolution of energy and then decrease prior to earthquake as seen previous analysis (Hayakawa and Ito, 1999). The earthquakes occurred in the study region are in very narrow range of magnitude (M 4.5-5.3) and their associated energies are also too close to incorporate the characteristics of energy in our analysis. However, the more distinguish characteristics of earthquakes such as epicentral distance, focal depth, and associated faults are correlated with enhancements or EM signatures to understand the fractal nature of pre-earthquake processes. Thus, present study restricted on moderate magnitude earthquakes occurred on different faults within relatively short time intervals. Hence, at few instances it becomes difficult to correlate these enhancements or EM signatures with occurrence of specific earthquakes; rather the sequence

of earthquakes would possibly decide the fractal nature, which may be deciphered based on a long time series of data. These limitations restrict us to relate the observed significant changes in geomagnetic signal with the magnitude and energy characteristics of earthquake and underlying processes.

**Comment 11.** Please one more time, introduce fractal dimensions in a proper way. This is important for readers. Further, multifractal study lead us to understand the spectra of fractal dimension. FDs have been used in various fields of studies. Some well-known references are: Philosophical Magazine Lett. 85(1) 33–40 (2005); Chaos, Solitons & Fractals 128, 71-82 (2019); Acta Biotheoretica, 52(1): 41-56 (2004); Journal of Thermal Stresses 44, 899-912 (2021); Geographical Analysis, 35(4): 310-328 (2003); Acta Mechanica 232, 1413-1424; (2021); Chaos, Solitons & Fractals, 35(1): 85-98 (2008); Advances in Space Research 74 (5), 2510-2529 (2024); Discrete Dynamics in Nature and Society, Vol. 2010, Article ID 194715, 22 pages (2010); Chaos, Solitons & Fractals, 45 (2): 115–124 (2012); Chaos, Solitons & Fractals, 49(1):47-60 (2013) Dynamics of Atmospheres and Oceans 106, 101459 (2024) A revision is required.

**Answer 11.** We appreciate the reviewer's comment suggesting more elaborate description of fractal dimensions. Accordingly, we have revised the Introduction section. We have first explained the fundamentals of the fractal method and later the common methods used in analysis of time series of geomagnetic signal. At the end, we explained advancement of fractal into spectra of fractal i.e. multifractal and the common multifractal methods used in recent study for analysis of geomagnetic signal. We have incorporated few relevant references from suggested list. (line number 90-173).